**W-band S/Z Relationships for Rimed Snow Particles: Observational Evidence from**
**Combined Airborne and Ground-based Observations**
Shelby Fuller [1], Sam Marlow [1], Samuel Haimov [1], Matthew Burkhart [1], Kevin Shaffer [1], Austin
Morgan [1], and Jefferson R. Snider [1,2]
[1] Department of Atmospheric Science, University of Wyoming, Laramie, WY, USA
[2] Corresponding Author, jsnider@uwyo.edu
**Abstract**
Values of undercatch-corrected liquid-equivalent snowfall rate (S) at a ground site and
microwave reflectivity (Z) retrieved using an airborne W-band radar were acquired during
overflights. The temperature at the ground site was between -6 and -15 $^o$C. At flight level, within
clouds containing ice and supercooled liquid water, the temperature was approximately 7 $^o$C
colder. Additionally, airborne measurements of snow particle imagery were acquired. The
images demonstrate that most of the snow particles were rimed, at least a flight level. A
relatively small set of S/Z pairs (4) are available from the overflights. Important distinctions
between these measurements and those of Pokharel and Vali (2011), who reported S/Z pairs and
an S/Z relationship for rimed snow particles, are 1) the fewer number of S/Z pairs, 2) the method
used to acquire S, and 3) the altitude, relative to ground, of the W-band Z retrievals. This
analysis corroborates that the S/Z relationship reported in Pokharel and Vali (2011) yields an S -
in scenarios with snowfall produced by riming - substantially larger than that derived using an
S/Z relationship developed for unrimed snow particles.
**1 - Introduction**
Improvement of methods used to measure snowfall and rainfall are an ongoing focus of
meteorological research. The various methods are ground-based instruments that evaluate the
mass of precipitation that falls into or onto a collector (precipitation gauges) (Brock and
Richardson 2001), ground-based radars (Wilson and Brandes 1979), and airborne and space-
borne radars (Matrosov 2007; Kulie and Bennartz 2009; Geerts et al. 2010; Skofronick-Jackson
et al. 2017). An objective of these approaches, whether used to make observations independent
of other methods (e.g., Kulie and Bennartz 2009), or as a component of multiple observations
(e.g., Cocks et al. 2016), is estimation of precipitation rate and accumulated precipitation
amount.
Many studies have investigated using radar for evaluating rainfall (for a review see
Wilson and Brandes 1979). There are two approaches. The first is research, both observational
and computational, that probes the relationship between rainfall rate (R) and radar-measured
values of range-corrected backscattered microwave power. The latter is commonly reported as an
equivalent radar reflectivity factor ($Z_e$). The second is operational in the sense that precipitation
gauges are used to calibrate measurements acquired using weather surveillance radars.
Complications associated with converting $Z_e$ to R, or converting a radar reflectivity factor[1] (Z) to
R,  can be grouped in four categories: 1) Inaccuracy in quantification of Z, 2) variation of the
R/Z relationship stemming from precipitation processes (e.g., coalescence and break up), 3)
difference between the volume of a radar range gate versus the much smaller volume of

---

[1] Radars are calibrated to report $Z_e$ (Smith 1984). Herein, radar reflectivities are reported as $Z = Z_e$ and as dBZ = $10\log_{10}(Z_e)$.

atmosphere sampled as precipitation falls to a gauge, and 4) vertical displacement between a
radar range gate and a calibrating gauge, especially at far ranges.
For situations with snowfall, methods employing either gauge or radar are associated
with complications beyond that incurred in rainfall (Matrosov 2007; Martinaitis et al. 2015;
Cocks et al. 2016). Problems associated with gauge measurements are wind-induced snow
particle undercatch, gauge capping, delayed registration, and blowing snow aliasing as snowfall.
Moreover, in a situation with snow particles more abundant within a radar range gate, compared
to rain drops, and where a measurement of Z is used to infer R via an R/Z relationship, the
resultant precipitation rate will likely be inaccurate. This is because hydrometeor shape, density,
and dielectric properties are all variable for snow particles while relatively invariant for rain
drops. Additionally, a snow particle's terminal fall speed varies with size (as is the case for
drops) and with particle shape and particle density. Going forward, we refer to the latter two
properties as shape and density.
The goals of this paper are as follows: 1) to describe measurements of undercatch-
corrected liquid-equivalent snowfall rate (S, mm h$^{-1}$) and how these were paired with W-band
measurements of reflectivity (Z, mm$^6$ m$^{-3}$) ; 2) to contrast the S/Z pairs against S/Z relationships
commonly applied in radar retrievals of S; and 3) to investigate why the S/Z pairs deviate from
predictions of some S/Z relationships.
In calculations of paired values of S and Z, density is an important parameter. Density is
commonly estimated using empirical data (e.g., Pokharel and Vali 2011, [PV11]). For graupel, a
snow particle that grows via collection of supercooled cloud droplets in a process commonly
referred to as riming, paired observations of particle mass and particle size have been used to
estimate density. There is considerable uncertainty in this approach. Based on data collected at
two northwestern US surface sites (Zikmunda and Vali 1972; Locatelli and Hobbs 1974), density
values differ by at least a factor of two at particle sizes smaller than 2000 μm (PV11; their Fig.
4). Given that the density of rime ice varies with droplet impact speed, droplet size, and
temperature (Macklin 1962), it is not surprising that the density-size relationships analyzed by
PV11 are so varied.

Table 1 and the following paragraphs overview W-band S/Z relationships applied in

instances with snow particles grown by vapor deposition (crystal), by collection of crystals
(aggregate snowflake), and by riming (rimed crystal and graupel). Henceforth, the latter two
snow particle types are collectively referred to as rimed snow particles.

In a computational study, Hiley et al. (2011) considered a variety of snow particle types

(column, plate, bullet rosette, sector plate, dendrite, and aggregate snowflake), employed a
parameterized ice particle size distribution (PSD) function (Field et al. 2005), accounted for a
range of temperature (-5 to -15 $^{o}$C) via the Field et al. parameterization, and developed a range of
S/Z relationships for snow particles. Except for the aggregate snowflakes (henceforth,
aggregates), the modeled particle types were vapor-grown crystals. Hiley et al.'s upper- and
lower-limit relationships are $S = 0.21 \cdot Z^{0.77}$ and $S = 0.024 \cdot Z^{0.91}$, respectively. Matrosov (2007)
developed an S/Z relationship for aggregates. In that work, parameterized PSDs from Braham
(1990) were employed, and a range of particle aspect ratios were factored into the calculations.
For aggregates, the S/Z relationship is $S = 0.056 \cdot Z^{1.2}$ (Matrosov 2007). It should be noted that
Hiley et al. (2011) and Matrosov (2007) employed similar, but not identical, computational
methods. Computational research was also conducted by Kulie and Bennartz (2009) who
adopted the wavelength-dependent density derived by Surussavadee and Staelin (2007) (200 kg
m$^{-3}$ at λ = 3.2 mm), modeled the snow particles as spheres, and applied PSDs based on Field et
al. The resultant S/Z is $S = 0.52 \cdot Z^{0.83}$ (Surussavadee and Staelin 2007; Kulie and Bennartz 2009;
henceforth, SSKB). Variance in the calculations discussed in this paragraph originate from
changes in density, shape, fall speed, PSD, and particle size as these changes are propagated
through the cloud-microphysical and microwave-scattering calculations.
In a hybrid approach (computational and an analysis of measurements), PV11 concluded
that most of the snow particles they imaged were rimed snow particles. Values of S were
calculated using a density-size function ($\rho_1$, discussed below), a fall speed-size function,
measured PSDs and measured particle images, and a determination of particle volumes. It was
assumed that a prolate spheroid approximated particle shape and that shape  was the basis for
determining a particle's sphere-equivalent volume and the particle's sphere-equivalent size. The
sphere-equivalent size was applied in the two functions. Values of Z were calculated using a
measured PSD, sphere-equivalent sizes, the $\rho_1$ function, and Mie Theory. PV11 presented
calculations of Z, obtained using two density-size relationships (their Eqs. 1 and 2) and compared
their calculated reflectivities to measurements of Z from a W-band radar. That led to their
conclusion that "...the lower density assumption…yielded closer correspondence to observed
reflectivities." Their recommendation for S as a function of measured Z - hereafter the S($\rho_1$)/Z
best-fit line - is $S = 0.39 \cdot Z^{0.58}$. Values of Z that were paired with the calculated values of S (i.e.,
the S/Z pairs from PV11 that we present in Sect. 4), and that were used to determine the S($\rho_1$)/Z
best-fit line, came from the WCR. In addition to variance in their values of S, coming from a
dependence on density, PV11 state that a value of S derived via their best-fit line is uncertain by
a factor-of-ten. That uncertainty is evident in the variance of S/Z data pairs about the S($\rho_1$)/Z line
in Fig. 11 of PV11. Those investigators, and Geerts et al. (2010), attributed the variance to use of
two-dimensional snow particle images in calculations of S and to actual variations of density,
shape, and particle size not accounted for in the calculations.
Another set of hybrid-type S/Z relationships was developed by Falconi et al. (2018; their
Table 2). These are based on measurements from a video disdrometer, weighing precipitation
gauge, microwave radiometer, and a vertically-pointing W-band radar. All these systems were
operated at the ground. The data set was stratified into intervals of lightly-rimed, moderately-
rimed, and heavily-rimed snow. A proxy for snow particle riming - radiometer measurements of
liquid water path – was the basis for the stratifications (von Lerber et al. 2017). The S/Z
relationships are $S = 0.10{\cdot}Z^{1.0}$ (lightly-rimed), $S = 0.079{\cdot}Z^{1.3}$ (moderately-rimed), and $S =$
$0.060{\cdot}Z^{1.4}$ (heavily-rimed).
Our focus is on surface measurements of S and on pairing of those measurements with
airborne measurements of Z. We also analyze airborne measurements of snow particle imagery.
The latter demonstrate that the particles observed at flight level were rimed. The imagery is the
basis for our assertion that our data set is relevant to ongoing investigations of using Z to
evaluate S in situations where precipitation is produced by riming.
Section 2 describes the setting of our study, the instruments we deployed, and recordings
we obtained using two data acquisition systems. One of the data systems was operated at a
ground site and the other on an aircraft. Section 3 is an analysis of the recordings; this section
also considers recordings from two additional, but ancillary, ground sites. Our findings are
discussed in Sect. 4 and summarized in Sect. 5. An Appendix (Sect. 6) explains how we
averaged recordings of near-surface W-band reflectivities and surface-based recordings of
snowfall.
Table 1 – W-band S/Z relationships from the literature, snow particle type, and values of minimum relative S difference

| Reference | Abbreviation used for reference | S/Z relationship | Snow Particle Type | Minimum relative S deference on December 15 2016 [a] | Minimum relative S difference on January 3 2017 [a] |
|---|---|---|---|---|---|
| Hiley et al. (2011) | - | $S=0.21 \cdot Z^{0.77}$ | Upper-limit S/Z relationship for vapor-grown crystals | 0.7 | 1.0 |
| Matrosov (2007) | - | $S=0.056 \cdot Z^{1.2}$ | Aggregates | 1.4 | 8.5 [b] |
| Surussavadee and Staelin (2007) and Kulie and Bennartz (2009) | SSKB | $S=0.52 \cdot Z^{0.83}$ | Spherical snow particles with density = 200 kg m$^{-3}$ | 0.3 | 0.2 [c] |
| Pokharel and Vali (2011) | PV11 | $S=0.39 \cdot Z^{0.58}$ | Rimed snow particles assuming the lower of two density-size relationships | 0.3 | 0.0 [d] |
| Falconi et al. (2018) | - | $S=0.060 \cdot Z^{1.4}$ | Snow particles classified as heavily rimed | 0.6 [e] | 8.5 |


[a] Minimum relative S difference is defined as the minimum of $|(S_{HP}-S)|/S$ where $S_{HP}$ is a measurement of undercatch-corrected liquid-equivalent
snowfall rate (Table 6) and S is a snowfall rate on an S/Z relationship line evaluated at one of the attenuation-corrected reflectivities (Sect. 4).
[b] Attenuation-corrected Z on this day (0.6 mm$^6$ m$^{-3}$) is smaller than the lower-limit Z (1 mm$^6$ m$^{-3}$) advised for this S/Z relationship (Matrosov
2007).

[c] Maximum relative S difference on this day is 0.4.
[d] Maximum relative S difference on this day is 0.7.
[e] Maximum relative S difference on this day is 0.9.

**2 - Site, Aircraft, and Instruments**
**2.1 - Site**

Analyzed herein are aircraft and ground data from 14/15 December 2016 and from 3

January 2017. The ground data were acquired in a forest/prairie ecotone on the eastern slope of
the Medicine Bow Mountains in southeast Wyoming (Figs. 1a-b). No ground-based observers
were deployed during the two snowfall events analyzed.

At one of three ground sites (HP in Figs. 1a-b) a hotplate precipitation gauge (Rasmussen

et al. 2011; Zelasko et al. 2018), a GPS receiver, and a data acquisition system were deployed.
Once per second, the data system ingested a hotplate-generated data string, combined that with
time-of-day from the GPS receiver [Coordinated Universal Time (UTC)], and recorded the
merged hotplate/UTC data string. The absolute accuracy of the time stamp is no worse than 2 s.

Overflights of the hotplate were done by the University of Wyoming King Air (WKA) on

14/15 December 2016 and on 3 January 2017. The flights were conducted in preparation for the
SNOWIE field project (Tessendorf et al. 2019) and were flown from the Laramie, WY Airport
(LA in Fig. 1a). Data acquisition on the WKA was also synchronized with UTC, but with much
better accuracy than at the hotplate.

Measurements of horizontal wind (speed and direction), temperature, relative humidity,

and pressure from the US-GLE AmeriFlux tower (AF in Figs. 1a-b) are also components the
analysis. The AmeriFlux data were provided to us as 30-minute averages (AmeriFlux 2021;
Marlow et al. 2023).


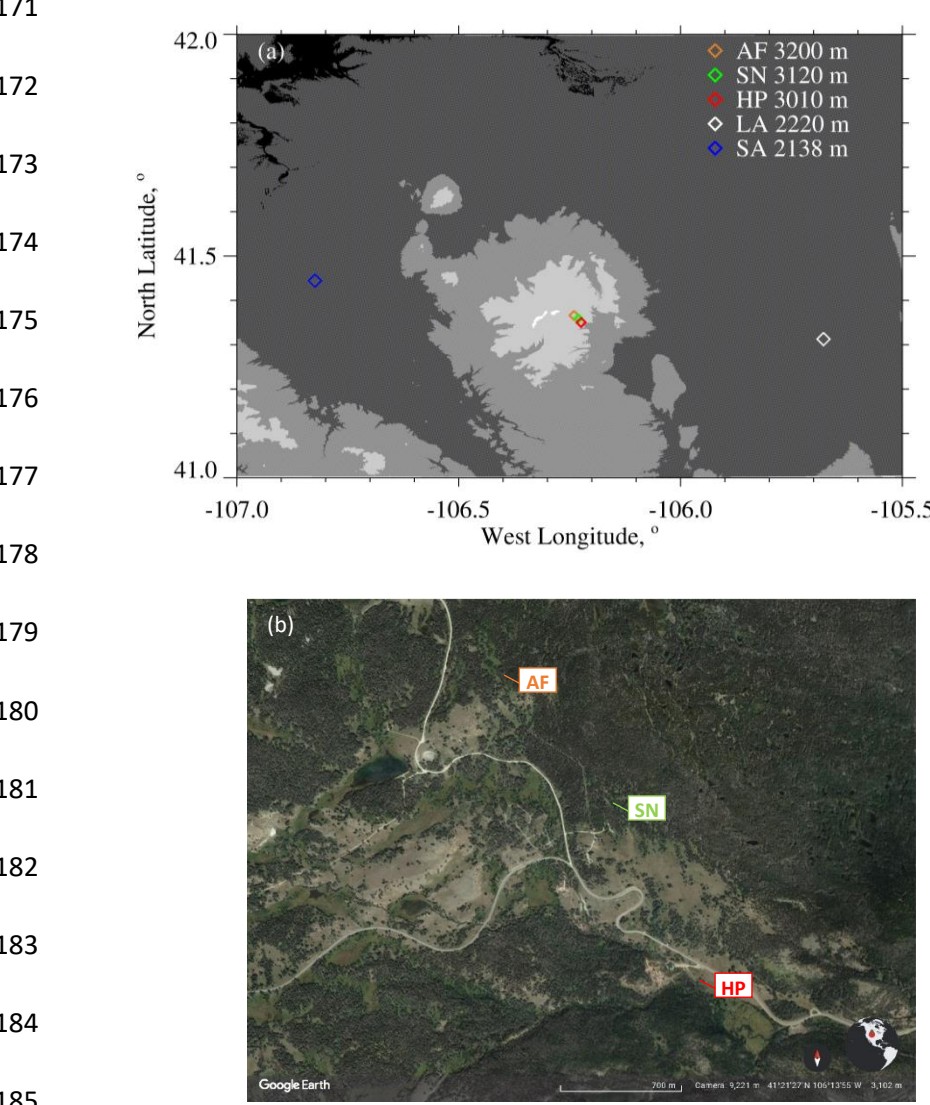
















Figure 1 – (a) Southeast Wyoming, airport at Saratoga, WY (SA), airport at Laramie, WY (LA),
and the ground sites: AF = US-GLE AmeriFlux tower, SN = Brooklyn Lake SNOTEL, and HP =
hotplate. Altitudes of the airports and ground sites are in the legend. Altitude thresholds for the
digital elevation map are 1500, 2000, 2500, 3000, and 3500 meters. (b) Close up of the AF, SN,
and HP ground sites (from © Google Earth).

**2.2 - University of Wyoming King Air (WKA)**

The following WKA measurements were analyzed: aircraft position, temperature, snow particle imagery, and three moments of the cloud droplet size distribution function. A Cloud Droplet Probe (CDP; Faber et al. 2018) was the basis for the droplet size distribution measurements and the derived moments. The latter are droplet concentration (N), cloud liquid water content (LWC), and mean droplet diameter (<D>). Snow particle imagery was obtained using a precipitation particle imaging probe (2DP; Korolev et al. 2011) and a cloud particle imaging probe (2DS; Lawson et al. 2006). These acquired two-dimensional images of particles between 200 to 6400 μm (2DP) and between 10 to 1280 μm (2DS).

**2.3 – The W-band Wyoming Cloud Radar (WCR)**

Retrievals from the up-looking and down-looking antennas of the WCR, operated on the WKA, were also analyzed. For this we used Level 2 WCR data[2] with reflectivities recorded as $dBZ = 10 \cdot \log_{10}(Z)$. The reflectivities were converted from dBZ to Z prior to processing. Additionally, values of the vertical-component Doppler velocity retrieved from below the WKA using the WCR's down-looking antenna were analyzed. The Doppler velocities were corrected for aircraft motion as described in Haimov and Rodi (2013). We use $V_D$ to symbolize the corrected vertical-component Doppler velocity and adopt the convention that $V_D > 0$ indicates upward hydrometeor motion.

---

[2] http://flights.uwyo.edu/uwka/wcr/projects/snowie17/PROCESSED_DATA/

212   The Level 2 WCR sampling was different on the two flight days and this difference is

213 shown in Table 2. Ground-based calibrations of the WCR's up-looking antenna and correlations

214 between in-flight retrievals acquired using the WCR's up-looking and down-looking antennas

215 were used to estimate the precision and absolute accuracy of the WCR-derived values of dBZ.

216 These are $\pm$ 1.0 dBZ and $\pm$ 2.5 dBZ, respectively (PV11).


Table 2 – Level 2 WCR sampling and the WKA overflight time

| Date | Level 2 WCR Vertical Sampling, m | Level 2 WCR Along-track Sampling, s | Overflight Time, UTC |
|---|---|---|---|
| 14/15 December 2016 | 23 | 0.23 | 00:00:38 (15 December 2016) |
| 3 January 2017 | 30 | 0.36 | 20:32:03 |


**2.4 - Hotplate Gauge**
Algorithms used to process hotplate measurements are described in Rasmussen et al.
(2011), Boudala et al. (2014), and Zelasko et al. (2018). Henceforth, these are referred to as R11,
B14, and Z18, respectively. This section describes how hotplate measurements acquired at the
HP site were analyzed. The hotplate deployed at the HP site is described in Wolfe and Snider
(2012), Z18, and in Marlow et al. (2023).
Five measurements fundamental to the steady state energy budget of the hotplate's
temperature-controlled up-viewing plate are output by the hotplate microprocessor as one-minute
running averages (Z18). These averages were merged with the GPS time and recorded at 1 Hz by
the data acquisition system (Sect. 2.1). With these measurements, calibration data (Marlow et al.
2023), and the algorithm developed by Z18, we calculated S in two steps. First, the five hotplate
measurements (electrical power supplied to the plate, ambient temperature, wind speed,
downwelling shortwave flux, and downwelling longwave flux) were input to Eq. 3 in Z18. The
output of that equation is a provisional liquid-equivalent precipitation rate. Second, the snow
particle catch efficiency, described in the next paragraph, was used to calculate S as the ratio of
the provisional rate and the catch efficiency.
Marlow et al. (2023; their Fig. 3b) report the relationship between snow particle catch
efficiency and wind speed that was applied in the calculation of S. There are three bases for this
relationship. First is the catch efficiencies R11 derived using measurements obtained from a
weighing gauge, operated within a double fence intercomparison reference shield, and collocated
measurements from an unshielded hotplate gauge. These paired measurements are symbolized
SRG (shielded reference gauge) and UHG (unshielded hotplate gauge). R11 plotted hotplate
catch efficiencies (i.e., UHG/SRG) versus wind speeds measured at 10 m AGL (their Fig. 8).
Second is Marlow et al.'s adjustment of R11's 10 m AGL wind speeds to 2 m AGL. The basis
for the adjustment is surface boundary layer parameters derived for R11's site (Kochendorfer et
al. 2018) and an equation from Panofsky and Dutton (1984; their Eq. 6.7). The adjustment was
made because the hotplate-derived wind speeds, both here and in Marlow et al. (2023), were
acquired at approximately 2 m above the snowpack surface. Third is Marlow et al.'s comparison
of SNOTEL-derived liquid-equivalent depth changes and hotplate-derived time-integrated
accumulations. The interval for the comparisons is 24 hours. Based on the comparison, which
has 57 paired values acquired at the sites labeled HP and SN in Fig. 1, the average fractional
absolute relative difference is 0.30. Marlow et al. also provided an estimate of the error in a
SNOTEL measurement (2.4 mm). At accumulation = 10 mm the error corresponds to a relative
error = 0.24. This indicates that SNOTEL contributed significantly to the SNOTEL/hotplate
variance and especially so for the smaller accumulations in Fig. 9a of Marlow et al. (2023).
Because of this, we do not limit the following estimate of hotplate precision to a subset of the 57
paired measurements. Based on our assessment of the average fractional absolute relative
difference, the hotplate precision applied in this analysis was taken to be 0.3.

The hotplate-derived wind speeds acquired at ~ 2 m, and discussed in the previous

paragraph, are henceforth symbolized $U_{PRO}$. The basis for these is a steady state energy budget of
the hotplate's temperature-controlled down-viewing plate and a proprietary algorithm (R11 and
Z18). The $U_{PRO}$ are reported by a hotplate as one-minute running averages (Z18) and we
recorded these at 1 Hz. Examples are the gray dots in Fig. 2. Additionally, we calculated and
analyzed one-minute-averaged values of $U_{PRO}$ and the corresponding standard deviations.
Examples of these are the black circles and the short vertical line segments in Fig. 2.


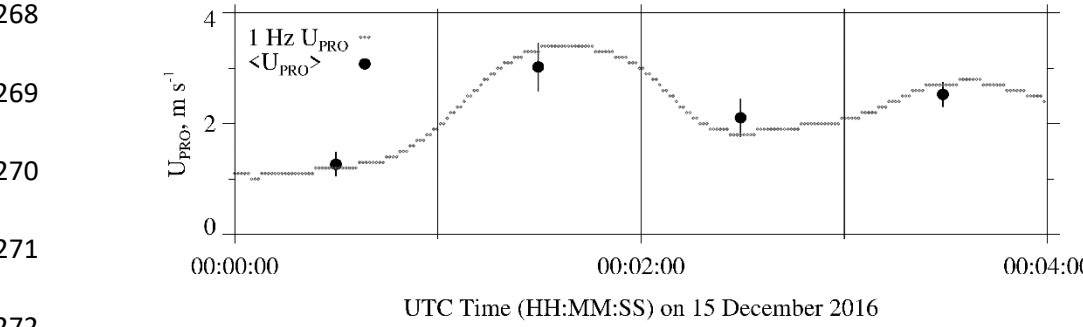



Figure 2 – Hotplate wind speed measurements ($U_{PRO}$) 00:00:00 to 00:04:00 on 15 December
2016. Gray dots are the one-minute running-average $U_{PRO}$ recorded at 1 Hz. Black circles are the
one-minute-averaged $U_{PRO}$ (± 1 standard deviation).

**3 - Analysis**
**3.1 - WKA Overflight Time**
The focus of our analysis is the two WKA flight segments shown in Figs. 3a-b. The maps
shown in the figures have the three ground sites (AF, SN, and HP) and the WKA flight tracks
(white line). The beginning-to-end time interval for the flight tracks is 100 s and these are
divided into ten 10-second intervals. The 10 s intervals are indicated with white diamonds.
Except for the turn evident in Fig. 3b, the flight tracks are straight, and the track direction is
approximately upwind to downwind.
Times that the WKA was closest to the HP site were evaluated by finding the point on the
flight track where the horizontal position of the WKA was closest to the hotplate's coordinates.
These times are symbolized $t_0$ and are referred to as overflight times. In Figs. 3a-b the downwind
end of the flight tracks end at the overflight time. The latitude/longitude position of the aircraft
was within 390 m of the hotplate at the overflight times. Table 2 has the overflight times on the
two flight days.








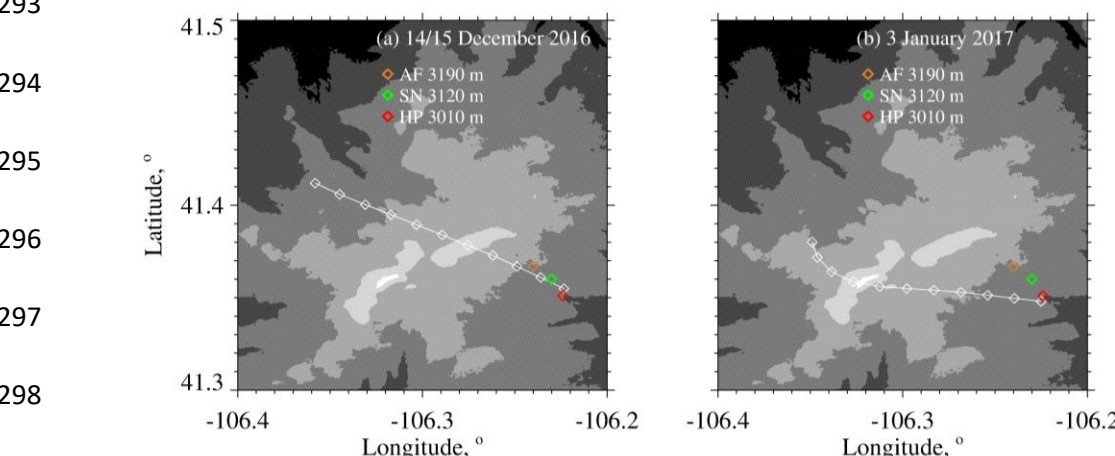

Figure 3 – (a) WKA flight track on 14/15 December 2016 for time interval = overflight time -
100 s to the overflight time. (b) WKA flight track on 3 January 2017 for time interval =
overflight time - 100 s to the overflight time. The white diamonds on the tracks are separated, in
time, by 10 s. Altitude thresholds for the digital elevation maps are 2600, 2800, 3000, 3200,
3400, and 3600 meters. Altitudes of the ground sites are in the legend.

**3.2 – Effect of Attenuation on WCR Reflectivities**

The presence of molecular oxygen, water vapor, cloud water, and snow particles within the WCR's transmission path will contribute to an attenuation of microwave intensity and will therefore negatively bias the retrieved reflectivities (Matrosov 2007; Hiley et al. 2011; Kneifel et al. 2015). Models of attenuation, radar remote sensing, and in situ measurements were used to calculate this bias. For oxygen, an attenuation coefficient from Ulaby et al. (1981; their Fig. 5.6), and temperature (T) and pressure (P) measurements from the AF (Table 3), were used. For vapor, an attenuation coefficient (Ulaby et al. 1981; their Eq. 5.22), and T, P, and relative humidity (RH) measurements from the AF (Table 3), were used. Concentrations of oxygen and water vapor and the oxygen and vapor path lengths are provided in Table 4. The latter is the vertical distance between the HP and the WKA. It was assumed that concentrations were uniform over this path length.

Attenuation by cloud water was derived using the WKA-measured T (Table 3), the WKA-measured LWC, path length (Table 4), and an attenuation formula (Liebe et al. 1989; Vali and Haimov 2001). The LWC applied in the formula is the maximum of CDP measurements acquired between $t_0$ - 10 s and $t_0$. This interval coincides with the interval the WCR's down-looking antenna was used to acquire reflectivities over the HP (Sect. 3.5). The path length for cloud water was derived as the vertical distance between cloud base [derived thermodynamically using AF measurements (Table 3)] and flight level. LWC was assumed uniform, at the maximum value, over the path length.

Snow particle mass concentration is typically reported as an ice water content (IWC, g m$^{-3}$) (Liu and Illingworth 2000). The contribution of IWC to attenuation was calculated using

measurements in Nemarich et. al (1988), who reported an attenuation coefficient equal to 0.9
dB/km per unit of IWC. Also used were retrievals of IWC acquired using the down-pointing
WCR antenna.  There are several steps in the calculation. First, all profiles of dBZ acquired
between $t_0$ - 10 s and $t_0$ were selected. Second, a maximum dBZ was selected at each of the
down-beam range gates (Table 2). Third, the dBZ maxima were increased by the overall two-
way attenuation in the final column of Table 4. Fourth, the profile of attenuation-corrected dBZ
was converted to a profile of attenuation-corrected Z. Fifth, a Z-to-IWC parameterization was
applied (IWC = $0.10 \cdot Z^{0.51}$; PV11; their Table 3). Sixth, the IWC profile was integrated, and the
derived ice water path was divided by the snow particle path length (Table 4). This calculation
produced a time- and range-averaged maximum IWC (Table 4). This IWC is the value applied in
the attenuation calculation.

Two-way attenuations ( $\Delta dB$ ), summed over contributions from the four components, are

presented in the final column of Table 4. Attenuation by snow and attenuation by liquid were the
most important components (> 50 %) on December 15 and January 3, respectively. Vapor
contributed 32 % to the overall on December 15, and the combination of vapor and snow
contributed 45 % on January 3. Equation 1 shows how an attenuation-corrected reflectivity ( $Z'$ )
was derived using an uncorrected reflectivity ( $Z$ ) and the $\Delta dB$ .

$$Z' = 10^{\left[\left[10 \cdot \log_{10}(Z) + \Delta dB\right]/10\right]}$$    (1)
Table 3 – Atmospheric state averages

| Date | WKA [a] Track Altitude, m | WKA [a] T, °C | AF [b] T, °C | AF [b] RH, % | WKA [a, c] Track Vector | WKA [a, c] Wind Vector | AF [b, c] Wind Vector |
|---|---|---|---|---|---|---|---|
| 14/15 December 2016 | 4546 | -13.9 | -6.3 | 86 | 310 / 130 | 274 / 32 | 250 / 8.5 |
| 3 January 2017 | 4196 | -21.7 | -14.6 | 77 | 280 / 120 | 265 / 27 | 260 / 5.4 |


[a] Altitude, temperature, track vector, and horizontal wind vector data obtained by averaging 1 Hz
WKA measurements. The averaging interval is 60 s and the interval starts at the overflight time,
minus 60 s, and ends at the overflight time.

[b] Temperature (T), relative humidity (RH), and horizontal wind vector data from sensors on the
US-GLE AmeriFlux tower (Sect. 2.1). The wind sensor was deployed at 26 m AGL (3223 m
MSL) and the T/RH sensor was deployed at 23 m AGL (3220 m MSL). The AF measurements
correspond to 30-minute averages closest to the overpass time. In the AF data set, time stamps on
the relevant AF recordings are 00:00 UTC (15 December 2016) and 20:30 UTC (3 January

2017).


[c] Vectors are presented in the following format: Direction of motion (degree relative to true
north) / speed (m s$^{-1}$).
Table 4 – Attenuating component concentration, one-way pathlength, and the overall two-way attenuation

| Date | Conc. Oxygen, kg m$^{-3}$ | Conc. Vapor, kg m$^{-3}$ | Maximum LWC, g m$^{-3}$ | Maximum IWC, g m$^{-3}$ | One-way Pathlength [a] Oxygen, Vapor, and Snow, km | One-way Pathlength [b] Cloud Water, km | Overall Two-way Attenuation, ΔdB |
|---|---|---|---|---|---|---|---|
| 15 December 2016 | 0.21 | 2.7x10$^{-3}$ | 0.01 | 0.27 | 1.54 | 1.09 | 1.41 [c] |
| 3 January 2017 | 0.21 | 1.3x10$^{-3}$ | 0.08 | 0.09 | 1.19 | 0.59 | 1.01 [d] |


[a] Vertical distance between HP and WKA

[b] Vertical distance between cloud base [derived thermodynamically using AF measurements (Table 3)] and WKA

[c] One-way attenuation coefficients are 0.03 dB/km for oxygen (Ulaby et al. 1981), 0.14 dB/km for vapor (Ulaby et al. 1981), 0.056
dB/km for cloud water (Liebe et al. 1989; Vali and Haimov 2001), and 0.24 dB/km for snow particles (Nemarich et. al 1988).

[d] One-way attenuation coefficients are 0.03 dB/km for oxygen (Ulaby et al. 1981), 0.073 dB/km for vapor (Ulaby et al. 1981), 0.49
dB/km for cloud water (Liebe et al. 1989; Vali and Haimov 2001), and 0.077 dB/km for snow particles (Nemarich et. al 1988).

**3.3 - Correction of Doppler Velocity**

We accounted for bias in $V_D$ (Sect. 2.3) due to deviation of the down-looking WCR

antenna from vertical. This was done by applying the correction described in Zaremba et al.
(2022) (their Eq. A4). The west-to-east and south-to-north particle velocities used in the
correction were assumed to be equal to component wind velocities. The latter were expressed as
linear functions of altitude using the information in the penultimate and last columns of Table 3.
The component velocities as functions of altitude and the linear equations relating velocity and
altitude are provided in the Appendix.
**3.4 - Hotplate Measurement of Wind Speed**

Here we compare the hotplate-derived wind speed to wind speed derived using an

R.M.Young rotating anemometer (R.M.Young 2001). The second of these is symbolized $U_{RMY}$
and the basis for the first ($U_{PRO}$) is a proprietary algorithm (Sect. 2.4). We are doing this
comparison because B14 showed that $U_{PRO}$ can be high-biased, relative to a conventional
anemometer, and because $U_{PRO}$ is the primary determinant of the rate that the up-viewing plate
dissipates sensible heat energy. Diagnosis of that heat transfer rate is our basis for calculating the
liquid-equivalent snowfall rate (Z18). The $U_{PRO}$ also determines the snow particle catch
efficiency and the latter was used in calculations of the undercatch-corrected liquid-equivalent
snowfall rate (Sect. 2.4).

The comparisons reported here were done at the Laramie, WY Airport in December

2019, and in January 2020. Compared to the HP site, the Laramie Airport site (indicated LA in
Fig. 1) is free of obstruction, out to 120 m, and experiences larger wind speeds. By mounting the
hotplate and the R.M.Young anemometer on rigid metal pipes, the hotplate's heated horizontal
surfaces (the up- and down-viewing plates seen in Fig. 1 of Z18) and the anemometer's spinning
axis (oriented horizontally) were both positioned at 2 m AGL. The pipes were separated
horizontally by 5 m. There was no precipitation on the days selected for the wind speed
comparisons. The values of $U_{PRO}$ and $U_{RMY}$ we analyzed were recorded with a data system that
time stamped the 1 Hz $U_{PRO}$ and 1 Hz $U_{RMY}$ with a relative timing accuracy no worse than 1 s.
A wind speed comparison - from 13 December 2019 - is shown in Fig. 4a. $U_{PRO}$ was
brought into the comparison by sampling it once per minute from files containing 1 Hz
recordings of the one-minute running-average $U_{PRO}$ (Sect. 2.4). $U_{RMY}$ was brought into the
comparison by starting with files containing 1 Hz recordings and converting these to one-minute
averages. Fig. 4a shows no evidence of bias and Fig. 4b demonstrates that the average absolute
departure between the $U_{PRO}$ and $U_{RMY}$ (both one-minute averages) is no larger than 1 m s$^{-1}$. Table
5 has eight more precipitation-free comparisons. Included in the table are temperature and wind
speed averaged over the comparison intervals (4 to 20 UTC), the slope of the linear-least-squares
fit line (forced through the origin, red line), and the lower and upper quartiles of the slope. The
quartiles were calculated using the method of Wolfe and Snider (2012). In contrast to Figs. 4a-b,
Figs. 4c-d make the comparison using 1 Hz values of $U_{PRO}$ and $U_{RMY}$. The larger scatter and
larger average absolute departure seen in these panels is a consequence of the hotplate's limited
time response, compared to the R.M.Young. We quantify the hotplate's response time in terms of
a calculated thermal response time. During wintertime at the Laramie Airport, and with wind
speed at 5 m s$^{-1}$, the down-viewing plate's thermal response time is approximately 60 s (results
not shown). Because the temperature of the down-viewing plate is actively controlled, this does
not translate to a 60 s lag between changes in wind speed and the hotplate response. The
$U_{PRO}/U_{RMY}$ departure is most evident at $U_{PRO} > 5$ m s$^{-1}$ (Fig. 4d) but this is not a concern for
U<sub>PRO</sub> on 14/15 December 2016 or on 3 January 2017. Snider (2023) demonstrated that the U<sub>PRO</sub>
was less than 5 m s$^{-1}$ at the hotplate during the two WKA overflights.

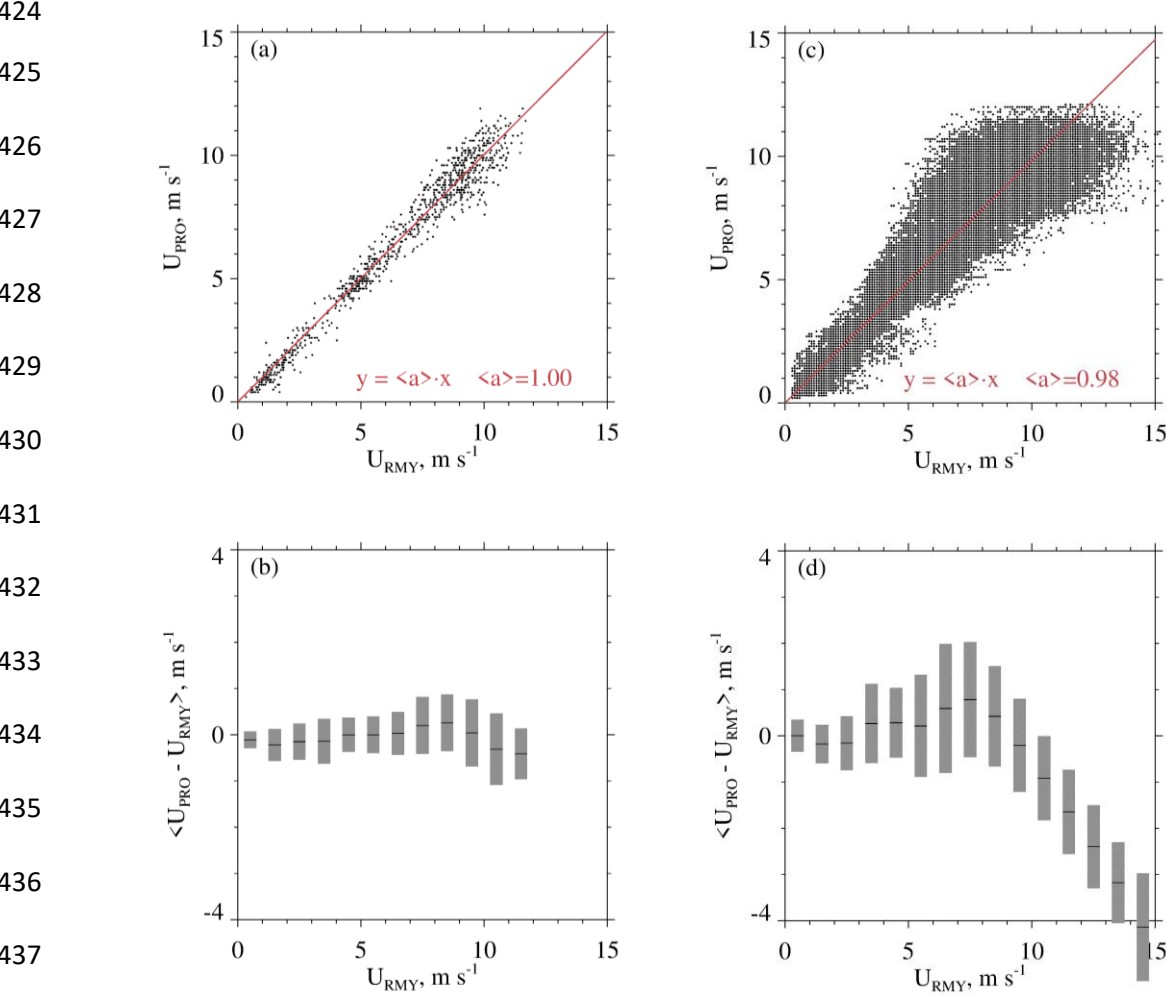

Figure 4 – (a) Scatterplot of one-minute-averaged $U_{PRO}$ and one-minute-averaged $U_{RMY}$.
Measurements were acquired at the Laramie, WY Airport 13 December 2019. The red line is a
linear-least-squares fit line (forced through the origin). (b) Average departure between one-
minute-averaged $U_{PRO}$ and one-minute-averaged $U_{RMY}$. Average departures were calculated for
discrete $U_{RMY}$ intervals, and the averages are indicated with short black horizontal lines. Gray
bars indicate ± 1 standard deviation. (c) Same as in (a) except for 1 Hz values of $U_{PRO}$ and $U_{RMY}$.
(d) Same as in (b) except for 1 Hz values of $U_{PRO}$ and $U_{RMY}$.


Table 5 - $U_{PRO}$ versus $U_{RMY}$ correlations

| Date, UTC [1] | $\langle T \rangle$ [2], ºC | $\langle U \rangle$ [2], m s$^{-1}$ | $\langle a \rangle$ [3] | a [4] First Quartile | a [4] Third Quartile |
|---|---|---|---|---|---|
| 7 December 2019 | -0.40 | 5.40 | 1.00 | 0.90 | 1.04 |
| 8 December 2019 | 2.70 | 4.10 | 0.99 | 0.90 | 1.04 |
| 10 December 2019 | -5.20 | 3.80 | 0.99 | 0.83 | 1.04 |
| 13 December 2019 | -1.50 | 6.60 | 1.00 | 0.93 | 1.06 |
| 18 December 2019 | -6.20 | 3.60 | 0.99 | 0.92 | 1.04 |
| 19 December 2019 | -6.90 | 2.70 | 0.95 | 0.84 | 0.99 |
| 6 January 2020 | -6.40 | 8.80 | 1.01 | 0.96 | 1.06 |
| 8 January 2020 | 0.30 | 4.20 | 1.00 | 0.87 | 1.05 |
| 11 January 2020 | -7.20 | 7.00 | 1.02 | 0.97 | 1.08 |


[1] Statistics presented are based on one-minute-averaged $U_{PRO}$ and one-minute-averaged $U_{RMY}$
measurements made between 04:00 to 20:00 UTC.

[2] Interval-averaged temperature and interval-averaged wind speed.

[3] Slope of the one-minute-averaged $U_{PRO}$ versus one-minute-averaged $U_{RMY}$ linear-least-squares
fit line, forced through the origin.

[4] Quartiles of the slope (see text)

**3.5 – Combined Aircraft and Surface Measurements**
Figure 5 has WCR and WKA measurements starting 100 s prior to $t_0$ and ending at $t_0$.
The sequences in Figs. 5a and 5c are reflectivities from both the up- and down-looking antennas.
In Fig. 5a the flight track (black dashed horizontal line) is at 4550 m and in Fig. 5c the flight
track is at 4200 m. At the $t_0$ in Fig. 5a, below the WKA, the maximum radar echo is +6 dBZ (Z =
4 mm$^6$ m$^{-3}$) and in Fig. 5c the maximum is -3 dBZ (Z = 0.5 mm$^6$ m$^{-3}$). Supercooled liquid water
was detected as the aircraft approached the ridgeline (Fig. 5b) and during the last 10 seconds of
the time sequence in Fig. 5d. During these encounters with supercooled liquid, the maximum
LWC values were 0.03x10$^{-3}$ and 0.08x10$^{-3}$ kg m$^{-3}$ on 14 December 2016 and 3 January 2017,
respectively. Values of N (Sect. 2.2) at times of maximal LWC were 3x10$^6$ and 100x10$^6$ m$^{-3}$ on
14 December 2016 and 3 January 2017, respectively. Even on 3 January 2017, the <D> (Sect.
2.2) associated with maximum LWC was sufficient for hexagonal plate crystals with diameter
larger than 100 μm to collide with the observed droplets with efficiencies > 0.1 (Wang and Ji

2000).

We temporally and spatially averaged the values of Z we compared with time-averaged
values of S. There are two reasons for this: 1) As discussed in Sect. 3.1, the WCR did not sample
Z exactly over the hotplate, and furthermore, the width of radar beam at 1500 m range - roughly
the distance between the aircraft and the ground at the overflight times - is 30 m and thus
considerably smaller than the minimum horizontal distance between the aircraft and the HP. 2)
Compared to the WCR, the hotplate is a relatively slow-response measurement system whose
output is commonly averaged over one-minute intervals (Z18).

FIGURE 5 WAS REVISED IN REVISION3.








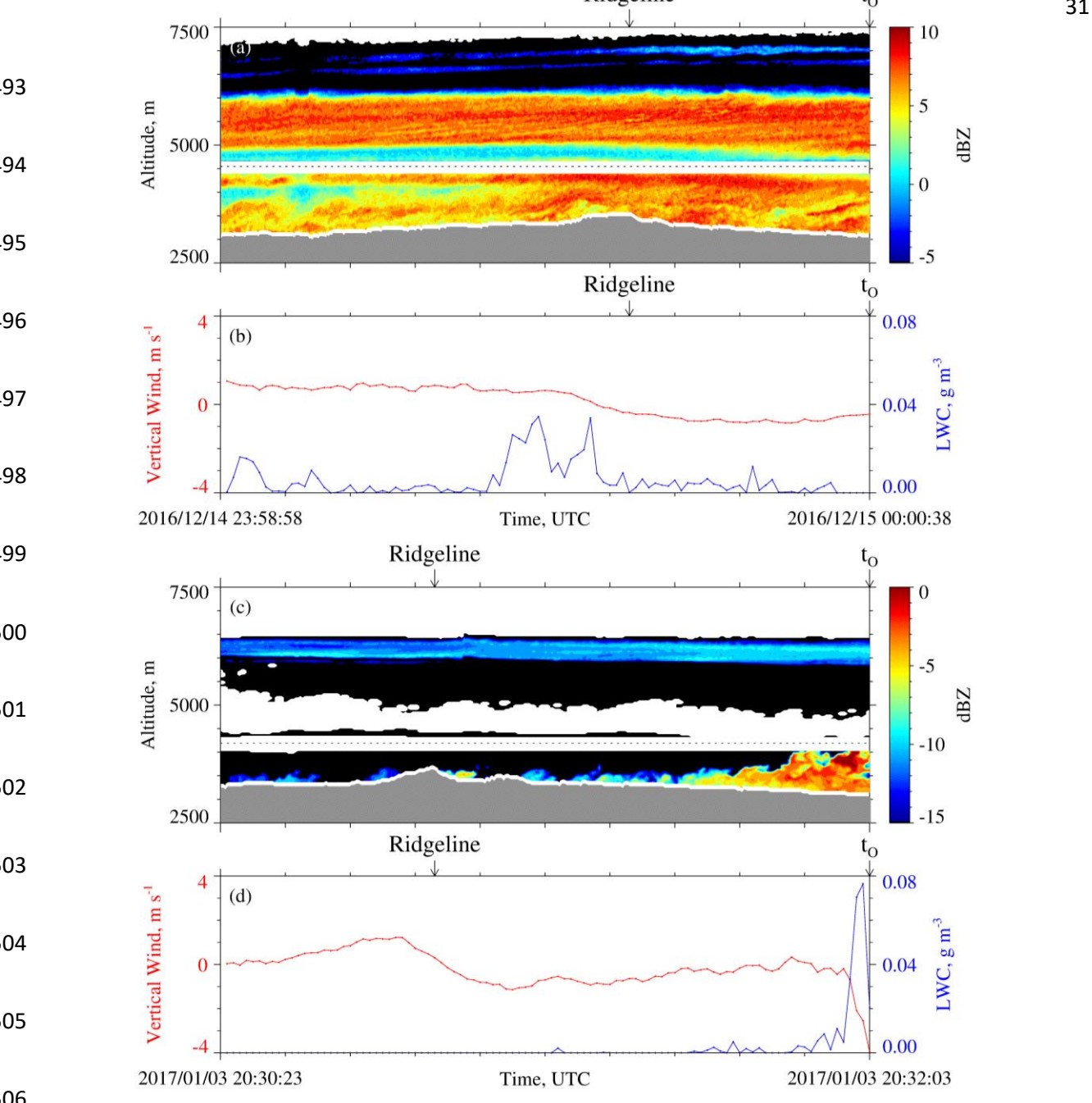















Figure 5 – (a) 100 s of WCR reflectivity and (b) 100 s of LWC and gust probe vertical
wind velocity ending at $t_0$ on 14/15 December 2016. (c) 100 s of WCR reflectivity and (d) 100 s
of LWC and gust probe vertical wind velocity ending at $t_0$ on 3 January 2017. In (a) and (c),
above and below the flight track, the roughly 200-m-deep WCR blind zone is evident,
reflectivity above (below) the flight track is from the up-looking (down-looking) WCR antenna,
black indicates dBZ values smaller than the minimum indicated in the color bar, white
immediately above the terrain indicates echo that was discarded because of ground clutter, and
white above the ground clutter and outside of the blind zone indicate dBZ < minimum detectable
signal.
In our analysis, the HP measurements were averaged over two adjacent 60 s intervals.
The first extends from $t_0$ to $t_0 + 60$ s (Fig. 6a) and the second from $t_0 + 60$ s to $t_0 + 120$ s (Fig.
6c). In Fig. 6a and in Fig. 6c, $t_{HP,B}$ symbolizes an interval's beginning time and $t_{HP,E}$ symbolizes
an interval's ending time. Formulas describing how these times were related to the beginning and
ending time of a corresponding WCR averaging interval are in the Appendix. Fig. 6b is a
schematic of the first WCR averaging interval and Fig. 6d is a schematic of the second. Again,
the subscripts "B" and "E" are used to indicate averaging beginning and ending times. Figures 6b
and 6d both have lines at the top of an averaging interval/domain. The slopes of these lines are
proportional to the ratio of two speeds. These speeds are a maximum likely snow particle speed
toward the ground ( $v_p$ ) and a horizontal wind advection speed ( $v_w$ ). The $v_p$ was calculated
using averaged vertical-component Doppler velocities and $v_w$ was calculated using a vertical
profile of horizontal winds, based on WKA horizontal wind measurements and AF horizontal
wind measurements (Figs. A1a-b), and using the WKA track vector (Table 3). An altitude ( $z' =$
3400 m) was assumed in the calculation of $v_w$ . This is the altitude of the ridges west and
northwest of the HP site (Figs. 3a-b). Picking the altitude to be either $z' = 3200$ m or $z' = 3600$ m
does not alter our findings.

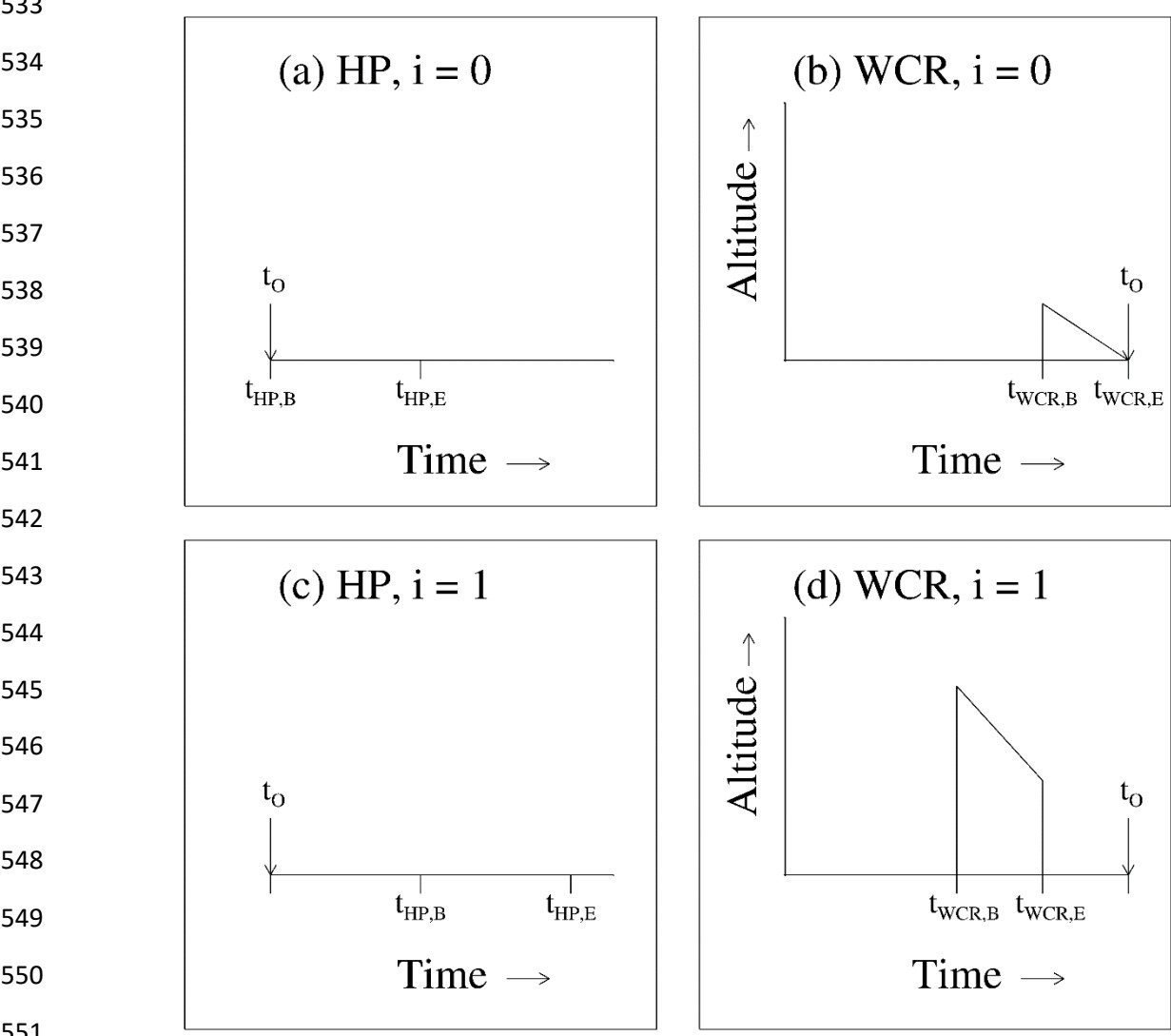

Figure 6 – (a and c) Representations of the $i = 0$ and $i = 1$ HP averaging intervals. (b and d)

Representations of the $i = 0$ and $i = 1$ WCR averaging intervals/domains. The $t_0$ is shown in all

panels. The subscripts "B" and "E" indicate beginning and ending times of the HP averaging

(panels a and c) and the beginning and ending times of the WCR averaging (panels b and d).


All panels in Fig. 6 are labeled with an index designating either the first averaging

interval ($i = 0$) or the second averaging interval ($i = 1$). Figures 7 and 8 present hotplate
snowfall measurements from 14/15 December 2016 and 3 January 2017. In these, and in
subsequent figures, colored circles surround the $i = 0$ and $i = 1$ indexes, blue is used to color-
code 15 December 2016, and red is used to color-code 3 January 2017. Additionally, Fig. 8 has
an $i = 2$ averaging interval. This is a special case discussed at the end of this section.

Figures 9a-b and Figs. 10a-b have enlarged views of the altitude-time WCR crossections

recorded on the two flight days. Different from Fig. 5a and Fig. 5c, these measurements are only
from the WCR's down-looking antenna. Additional differences are the following: 1) The plots
are set up so that Z and $V_D$ structures downwind of the hotplate can be seen. These structures are
discussed in the following section. 2) The WCR measurements are shown for 50 s of flight. With
the WKA ground speed approximately 125 m s$^{-1}$ (Table 3), the distance along the abscissa is
6250 m. 3) Colored circles that surround the indexes are placed below the WCR averaging
intervals/domains. The latter are drawn with solid black lines and are seen to overlay both the Z
and $V_D$ altitude-time crossections. Consistent with Figs. 6b and 6d, and the Appendix, one of
these black lines is vertical and another is negatively sloped. Figs. 10a-b also have the $i = 2$
intervals/domains discussed at the end of this section.

THIS FIGURE WAS REVISED IN REVISION3.











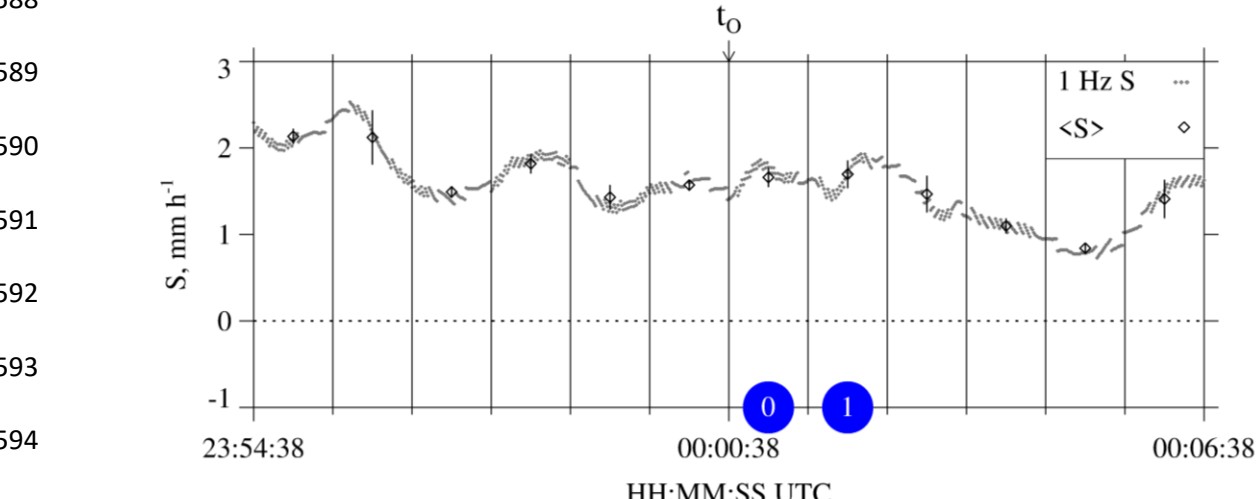









Figure 7 – Twelve minutes of HP snowfall measurements from 14/15 December 2016. Gray dots
are S values calculated using hotplate output recorded at 1 Hz. Black diamonds are the one-
minute-averaged values ($\pm$ 1 standard deviation). The $t_0$ is shown above the panel and blue
circles designate the $i = 0$ and $i = 1$ HP averaging intervals.

THIS FIGURE WAS REVISED IN REVISION3.

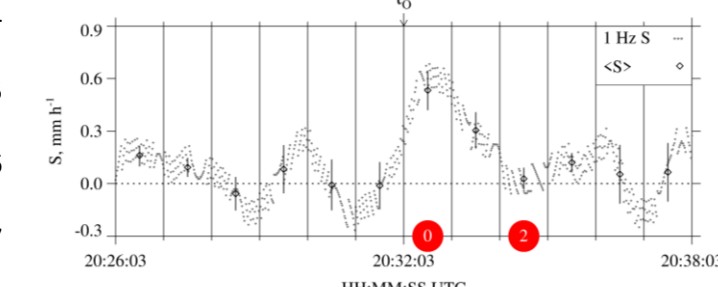










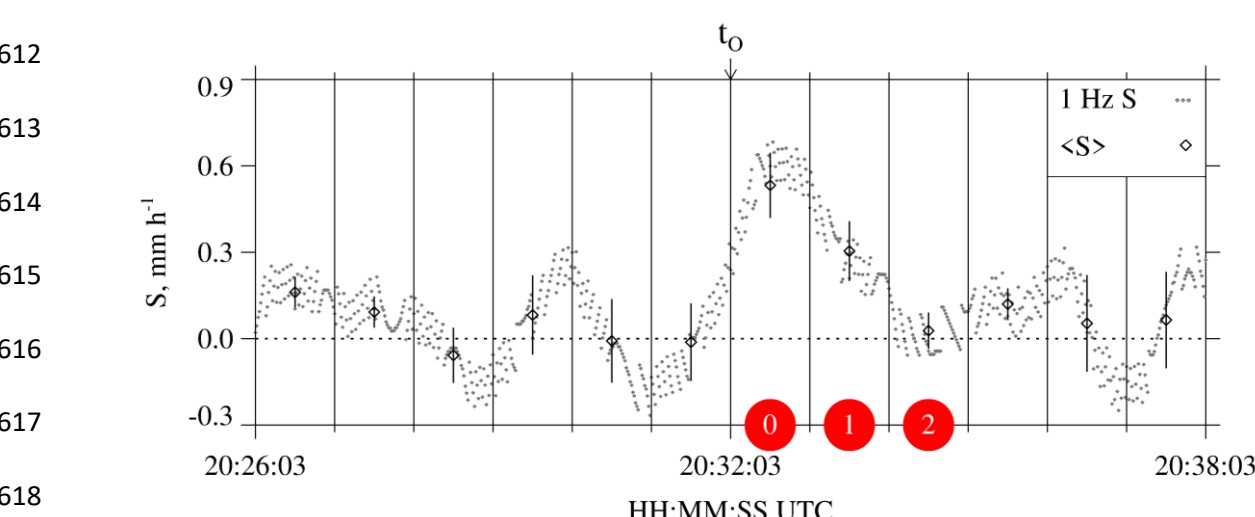


Figure 8 – Twelve minutes of HP snowfall measurements from 3 January 2017. Gray dots are S
values calculated using hotplate output recorded at 1 Hz. Black diamonds are the one-minute-
averaged values ($\pm$ 1 standard deviation). The $t_0$ is shown above the panel and red circles
designate the $i = 0$, $i = 1$, and $i = 2$ HP averaging intervals. The $i = 2$ interval is a special case
discussed at the end of Sect. 3.5.

THIS FIGURE WAS REVISED IN REVISION3.

















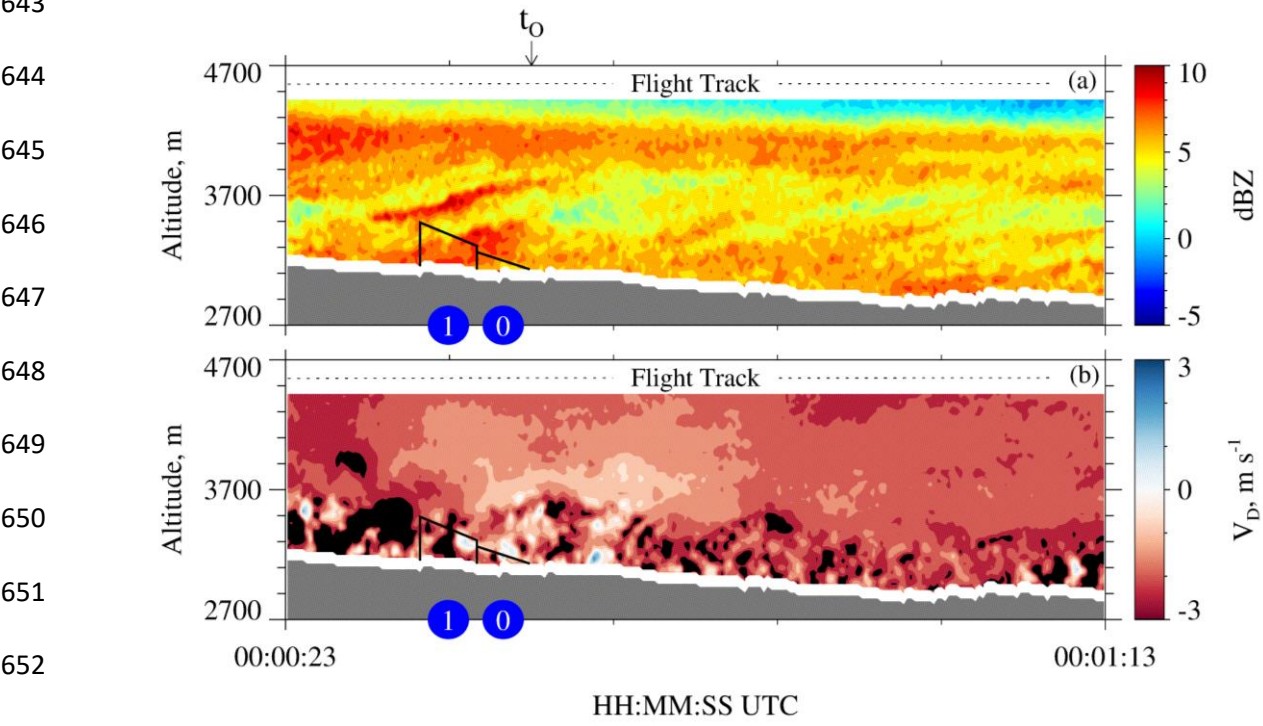












Figure 9 – 50 s of measurements from the down-looking WCR antenna on 15 December 2016.
(a) Crossection of reflectivity $t_0$ - 15 s to $t_0$ + 35 s. (b) Crossection of Doppler velocity $t_0$ - 15 s to
$t_0$ + 35 s. The $t_0$ is shown above the top panel. In both panels, the solid black lines (vertical and
sloped) encompass the $i = 0$ and $i = 1$ WCR averaging intervals/domains and blue circles
designate the WCR averaging intervals.

THIS FIGURE WAS REVISED IN REVISION3.

















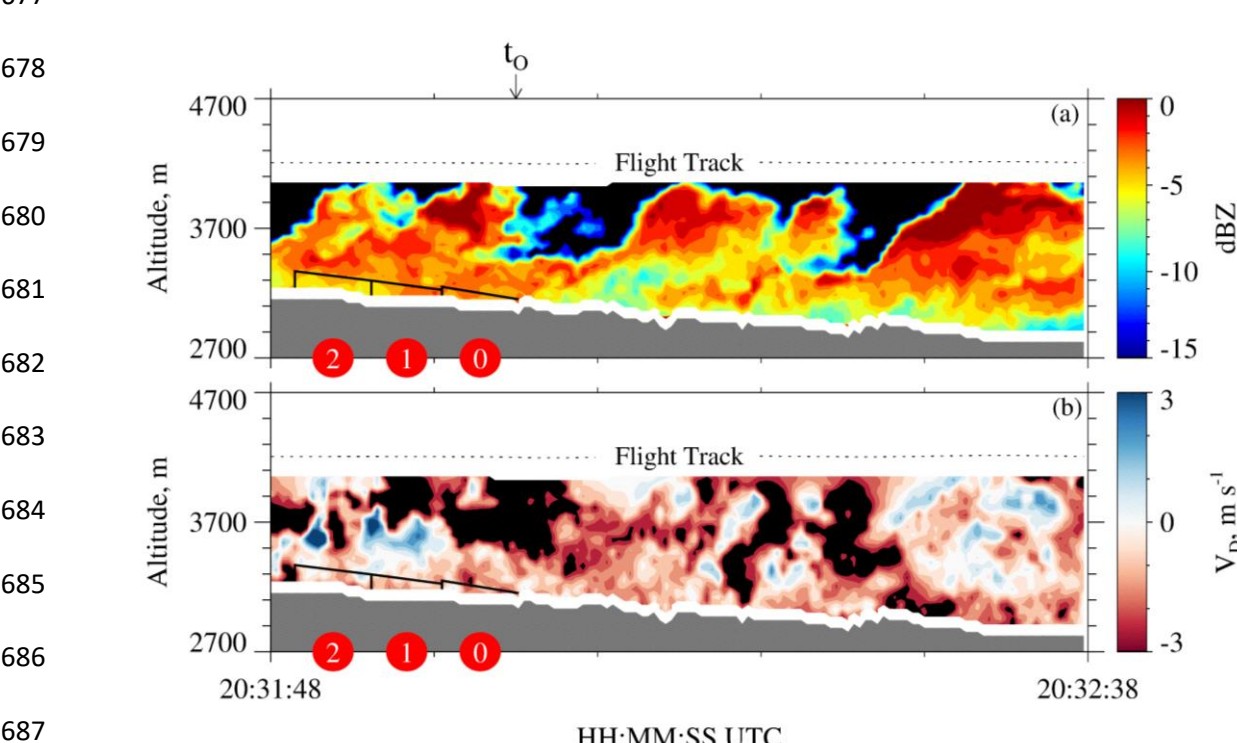

Figure 10 – 50 s of measurements from the down-looking WCR antenna on 3 January 2017. (a)
Crossection of reflectivity $t_0$ - 15 s to $t_0$ + 35 s. (b) Crossection of Doppler velocity $t_0$ - 15 s to $t_0$
+ 35 s. The $t_0$ is shown above the top panel. In both panels, the solid black lines (vertical and
sloped) encompass the $i = 0$, $i = 1$, and $i = 2$ WCR averaging intervals/domains and red circles
designate the $i = 0$, $i = 1$, and $i = 2$ WCR averaging intervals/domains. The $i = 2$ interval/domain
is a special case discussed at the end of Sect. 3.5.

697   The $i = 0$ and $i = 1$ averages of S and Z are presented in Table 6 and the corresponding

698 averaging intervals are viewable in Fig. 7 and Fig. 9a (15 December 2016) and in Fig. 8 and Fig.

699 10a (3 January 2017). According to the averaging scheme (Fig. 6), the $i = 1$ HP averaging

700 interval is time-shifted positively compared to the $i = 0$ HP averaging interval and the $i = 1$

701 WCR averaging interval is time-shifted negatively compared of the $i = 0$ WCR averaging

702 interval. This arrangement of the averaging intervals is one way to average while also accounting

703 for wind advection of the snow particles.

704   As discussed earlier in this section, the averaging scheme initializes with 60-second

705 blocks of HP data between $t_O$ and $t_O + 120$ s. When we applied the scheme to data from 3 January

706 2017, but outside the specified time range, an inconsistency was documented. This is apparent in

707 Fig. 8, where the $t_O + 120$ s to $t_O + 180$ s interval (i.e., the $i = 2$ interval) has negligible average S,

708 while in Fig. 10, the $i = 2$ interval has a non-negligible average Z ($\sim 0.3$ mm$^6$ m$^{-3}$). A firm

709 explanation is not available for the inconsistency, but a factor may be the convective nature of

710 the fields in Figs. 10a-b. Because of the inconsistency, only averages corresponding to the $i = 0$

711 and $i = 1$ intervals are analyzed further.

Table 6 – Average wind measurements, average hotplate measurements, average WCR measurements, and attenuation-corrected
reflectivities

| Date | $v_w$ [a], m s$^{-1}$ | i index | $S_{HP} \pm \sigma$ [b], mm h$^{-1}$ | WCR Samples [c] | $<V_D>$ [d], m s$^{-1}$ | $\sigma_{V_D}$ [e], m s$^{-1}$ | $v_p$ [f], m s$^{-1}$ | $<Z> \pm \sigma_Z$ [g], mm$^6$ m$^{-3}$ | $Z'$ [h], mm$^6$ m$^{-3}$ |
|---|---|---|---|---|---|---|---|---|---|
| 15 December 2016 | 7.4 | 0 | 1.7±0.1 | 42 | -1.3 | 0.9 | 2.2 | 4.9±2.1 | 6.8 |
| 15 December 2016 | 7.4 | 1 | 1.7±0.2 | 149 | -1.8 | 1.2 | 3.0 | 5.6±1.1 | 7.8 |
| 3 January 2017 | 8.9 | 0 | 0.5±0.1 | 22 | -0.9 | 0.8 | 1.7 | 0.49±0.05 | 0.62 |
| 3 January 2017 | 8.9 | 1 | 0.3±0.1 | 35 | -0.8 | 0.4 | 1.2 | 0.50±0.10 | 0.63 |


[a] Horizontal wind advection speed (Eq. A7) calculated using values from the penultimate and last columns of Table 3.
[b] One-minute average of the undercatch-corrected liquid-equivalent snowfall rate (± 1 standard deviation). An example averaging
interval is the $i = 0$ interval in Fig. 7.
[c] Number of samples used to calculate the WCR statistics. The averaging intervals/domains (e.g., $i = 0$ in Figs. 9a-b and in Figs. 10a-
b) encompass the WCR samples which are the basis for the WCR statistics presented in this table.
[d] Average of Doppler velocity within the averaging intervals/domains.
[e] Standard deviation of Doppler velocity within the averaging intervals/domains.
[f] Maximum likely snow particle speed toward the ground (Eq. A8).
[g] Average reflectivity (± 1 standard deviation). These values are not corrected for attenuation.
[h] Attenuation-corrected reflectivities. These were derived using reflectivities from the penultimate column of this table, attenuations
from Table 4, and Eq. 1.

### 3.6 - Snow Particle Imagery


In Fig. 9a and Fig. 10a, the time for a snow particle to move the abscissa and ordinate
distances is different. The ratio of these two times is 2.6. This follows from our choice of
abscissa and ordinate ranges, from values of particle fall speed (1 m s$^{-1}$) and horizontal wind
advection speed (8 m s$^{-1}$), which we assumed, and from the WKA ground speed ( $gs \sim 125$ m s$^{-1}$;
Table 3). The assumed values are approximately consistent with values of $<V_D>$ and $v_w$, in
Table 6, and with the $V_D$ sign convention (Sect. 2.3). We also used $gs = 125$ m s$^{-1}$ to scale
(virtually) the time axes in Fig. 9a and Fig. 10a to a horizontal distance. Within the scaled
coordinate frames, we assumed that all snow particle trajectories have negative slope ($\Delta z / \Delta x = $ -
1 m s$^{-1}$ / 8 m s$^{-1}$ = -0.12) and that all trajectories are stationary. However, both assumptions seem
inconsistent with the reflectivity structures in Fig. 5a, where positively-sloped particle fall
streaks are evident at $\sim$ 5500 m, inconsistent with Fig. 9a where positively-sloped fall streaks are
at $\sim$ 3500 m, and inconsistent with the positively-sloped fall streaks in Fig. 10a. On both flight
days, the fall streaks evince particle sources that move horizontally and with a horizontal speed
that is larger than the $v_w = 8$ m s$^{-1}$ applied in the estimate of the trajectory slope. It may be that
the source's horizontal speed is comparable to the flight-level WKA-derived horizontal wind (27
to 32 m s$^{-1}$; Table 3) but we do not have data needed to verify that assertion. Based on the
assumption that snow particles followed the fall streaks while both were advecting horizontally,
we looked *downwind* of the hotplate - at a time later than $t_0$ in Fig. 9a and Fig. 10a - for particles
that became those that produced snowfall at the hotplate.
Particle images from 15 December 2016 were analyzed using the 2DP. With this
instrument the maximum all-in particle size (in the horizontal direction perpendicular to flight) is
6400 μm and the particle size resolution is 200 μm (Sect. 2.2). Within the time interval picked
for this analysis (discussed below), particles sizing in the smaller of the two spectral modes, with
mode size ~ 400 μm, were more numerous (results not shown). Because the 400 μm particles are
poorly resolved by the 2DP, and the same can be said for somewhat larger particles, those
smaller than 1000 μm were excluded from the following analysis. Figure 11a shows imagery
from 12 s of measurements acquired near the end of the sequence in Fig. 9a (00:01:02 to
00:01:14). This time interval was selected by tracing forward from $t_0$, along the slope of the fall
streaks, to the flight level. Many of the particles are rounded (indicating riming) and a few have
arms likely due to incomplete conversion of branched crystals to rimed snow particles. The mode
size corresponding to these images is 1600 μm. No liquid water was detected with these particles
(LWC < 0.01x10$^{-3}$ kg m$^{-3}$; Fuller 2020; her Fig. 8), but liquid was detected, at ~ 00:00:00, as the
aircraft approached the ridgeline (Figs. 5a-b).

Turning to imagery from 3 January 2017, the most appropriate location for analysis

would be through the second billow structure evident in Fig. 10a (i.e., very close to the middle of
the Fig. 10a sequence). This billow sourced a fall streak that terminated at the hotplate (i.e., at
the time $t_0$ indicated in the figure). However, the aircraft only clipped the top of this billow, and
it was only when sampling the billow seen ~ 13 s earlier that larger ice particle concentrations (~
20,000 m$^{-3}$) (Fuller 2020; her Fig. 10) and larger LWC (~ 0.08x10$^{-3}$ kg m$^{-3}$; Fig. 5d) were
detected. Maximum reflectivities were the same in all three billows (Z ~ 1 mm$^6$ m$^{-3}$; 0 dBZ), so it
was assumed that imagery collected in the first billow (20:32:00 to 20:32:02) was representative
of what was falling toward the hotplate. The 2DS was used to image these particles (Fig. 11b);
with this instrument the maximum all-in particle size (in the horizontal direction perpendicular to
flight) is 1280 μm and the size resolution is 10 μm (Sect. 2.2). Most of the objects in Fig. 11b
appear to be rimed and their mode size is ~ 400 μm. It is also noted that particles smaller than
100 μm were eliminated from these images, however, compared to the ~ 400 μm particles those
smaller than 100 μm were significantly less abundant (results not shown).

(a)

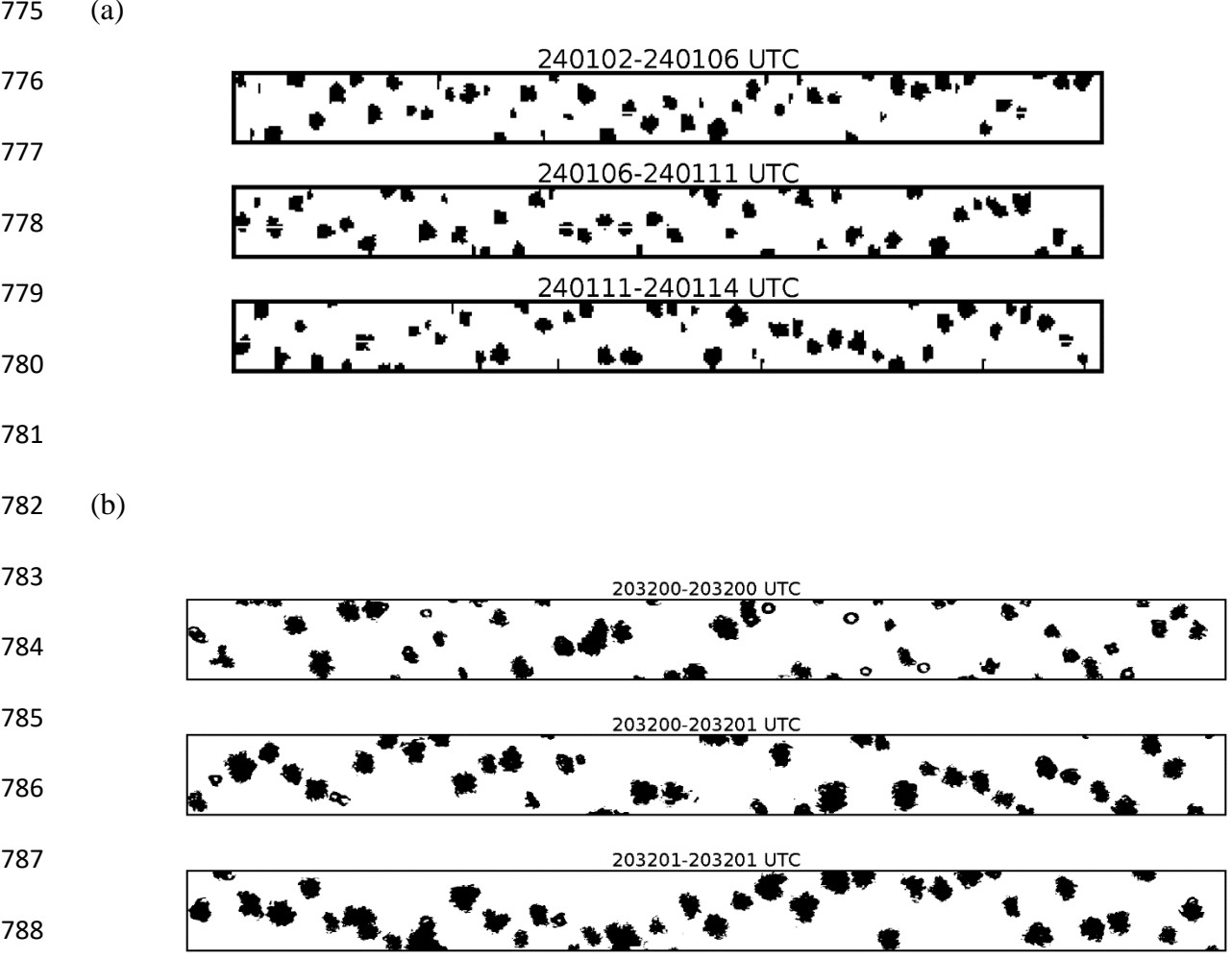

(b)

Figure 11 – (a) 2DP particle imagery from 15 December 2016. The height of the strips is 6400
μm. These particles are estimated to be representative of those that fell from flight level toward
the hotplate. (b) 2DS particle imagery from 3 January 2017. The height of the strips is 1280 μm.
These particles are estimated to be representative of those that fell from flight level toward the
hotplate.

**3.7 – S/Z Relationships**
Our S/Z pairs are presented in Table 6 where the indexes ($i = 0$ and $i = 1$) are used to
indicate results derived for the averaging intervals. In the penultimate column of Table 6,
reflectivities are not corrected for attenuation, however, in the last column of Table 6 and in Fig.
12, the attenuation-corrected reflectivities are presented. Reflectivities from the penultimate
column of Table 6, attenuations from Table 4, and Eq. 1 were used to calculate the corrected
reflectivities. Also shown in Fig. 12 (black filled circles) is a subset of the S/Z pairs from PV11's
Fig. 11 ($0.01 < Z < 10$ mm$^6$ mm$^{-3}$) and the PV11 best-fit line (black). Results from PV11 are
specified as S($\rho_1$)/Z because those authors applied the lower of two density-size functions ($\rho_1$),
and the lower of two fall speed-size functions, with airborne measurements, in calculations of
snowfall rates (Sect. 1 and Table 1).
THIS FIGURE WAS REVISED IN REVISION3.






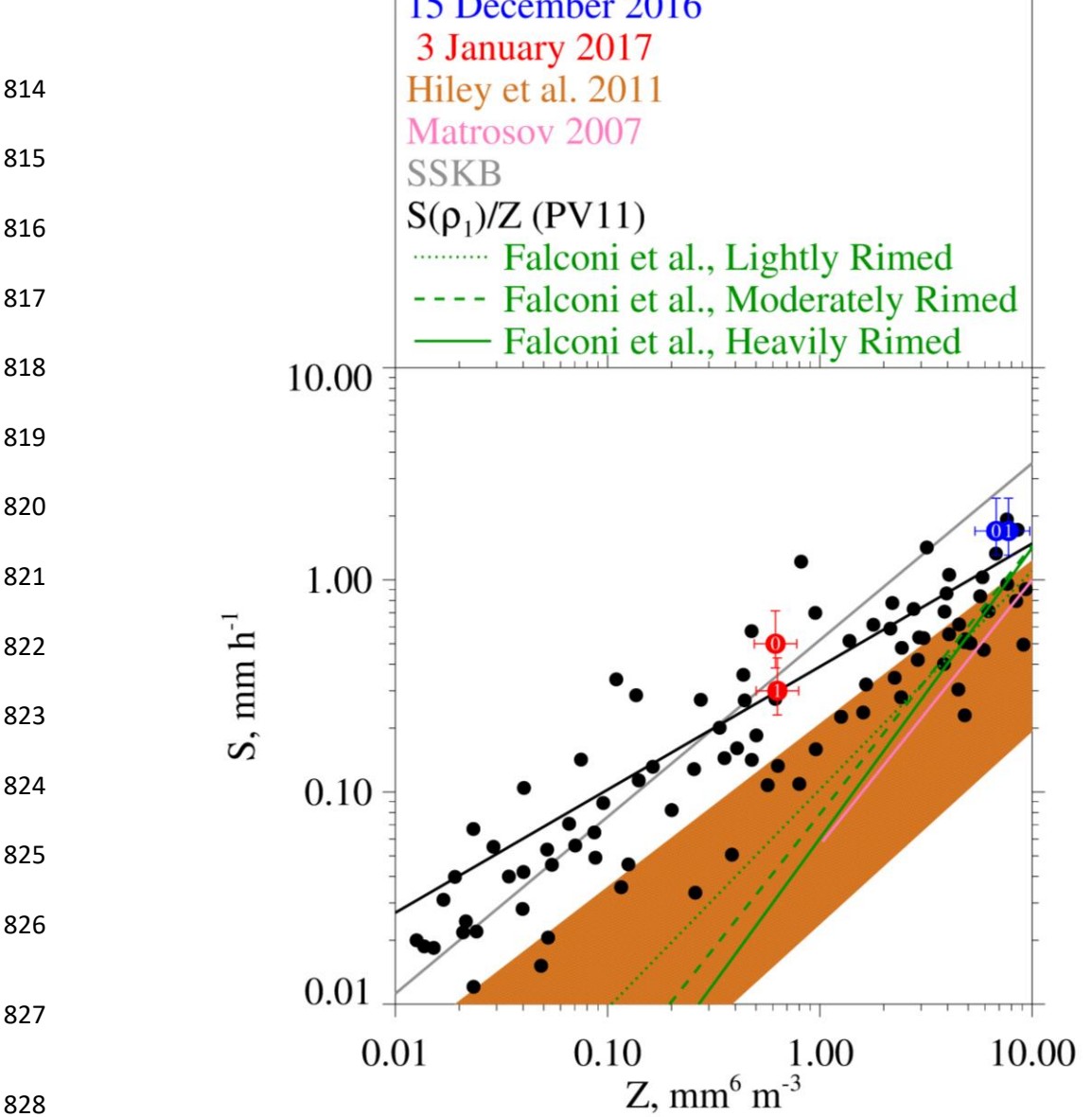

Figure 12 – Snowfall rate versus radar reflectivity. Red and blue circles are plotted at

attenuation-corrected reflectivities (Table 6) for the $i = 0$ and $i = 1$ averaging intervals. Error

bars on these points represent precisions of the reflectivity (Sect. 2.3) and snowfall rate (Sect.

2.4) measurements. Also plotted are the S/Z relationship lines from Sect. 1 and Table 1. These

are the S/Z lines defining the swath of S/Z relationships from Hiley et al. (2011), the S/Z

relationship from Matrosov (2007), the S/Z relationship abbreviated SSKB, PV11's best-fit line,

and the S/Z relationships from Falconi et al. (2018) (their Table 2). The $S(\rho_1)/Z$ points (black

filled circles) are a subset from PV11's Fig. 11 ($0.01 < Z < 10$ mm$^6$ mm$^{-3}$).

There are two potential biases in the values of snowfall rate we tabulate (Table 6) and
plot (Fig. 12). First, the two snowfall events had flight-level vertical wind velocities (Figs. 5b
and 5d) that were positive (upward) upwind of the ridgeline, and vice versa downwind of the
ridgeline. Except for the strongest downdraft on 3 January 2017, the magnitude of this variance
is ~ 1 m s$^{-1}$ (Figs. 5b and 5d). Assuming 1 m s$^{-1}$ was the downward wind immediately over the
hotplate, the snow particles would have approached the HP gauge faster than their fall speed.
Our basis for stating this is fall speeds for the mode sizes discussed in Sect. 3.6 (1600 and 400
μm) and our assumption that the particles were graupel. (Table 7 has these characteristic sizes
and fall speeds.) However, the conjectured downdraft speed is likely an overestimate - because
of divergence occurring as the draft approached the surface - and because the sizes in Table 7
likely underestimate what fell to the hotplate. Relevant to the last of these assertions, we used the
altitude/T/RH measurements (Table 3) to calculate the vertical distance available for growth via
riming, and thus for a fall speed increase, between the flight level and the lifted condensation
level. Assuming an adiabatically-stratified liquid cloud and unit collection efficiency (these
assumptions overestimate growth by riming), and no change of particle crossection
(underestimates growth by riming), our calculations indicate that relative increases of size and
fall speed were 40 and 20 %, respectively, on 3 January 2017, and that these relative increases
were a factor-of-two larger on 15 December 2016.

Table 7 – Estimates of snow particle fall speed

| Date | Mode Size, μm | Assumed Particle Type | Fall Speed, m s$^{-1}$ | Reference |
|---|---|---|---|---|
| 15 December 2016 | 1600 | graupel | 1.4 | PV11; assuming $\rho_1$ in their Fig. 5 |
| 3 January 2017 | 400 | graupel | 0.7 | PV11; assuming $\rho_1$ in their Fig. 5 |


Second, there is concern that values of S from 3 January 2017 are underestimated.
Although values of S must be > 0, we presented 1 Hz values (gray points, Fig. 8) approaching -
0.3 mm h$^{-1}$. Negative values resulted because we did not impose a threshold of 0 mm h$^{-1}$ on the
uncorrected snowfall rates (this thresholding is discussed in Z18) and because negative snowfall
rate values (uncorrected for catch inefficiency) are amplified by the gauge-catch correction (Sect.
2.4). The implication is that 0.2 mm h$^{-1}$ could be added to the one-minute averaged values of
snowfall rate in Table 6 and in Fig. 12. Here, the assumption is that an averaged S of -0.2 mm h$^-$
$^1$, in Fig. 8, indicates no snowfall at the hotplate; however, because the hotplate was operated
autonomously (Sect. 2.1) we have no way to verify the assumption.

**4 – Results**

Figure 12 shows our four snowfall rate/reflectivity pairs (red and blue circles) after the

reflectivities were corrected for attenuation. The error bars on these data pairs represent the
precision of the Z measurement (Sect. 2.3) and the precision of the S measurement (Sect. 2.4).
Presentation clarity was what guided the selection of S and Z axis ranges in this figure but with
the consequence that 32 of PV11's S/Z pairs are not shown because they plot at $Z > 10$ mm$^6$ m$^{-3}$.
The way that the PV11 data pairs scatter closest to $Z = 10$ mm$^6$ m$^{-3}$, combined with the fact that
the PV11 data pairs at $Z > 10$ mm$^6$ m$^{-3}$ are not shown, could lead to the interpretation that the
slope describing the best-fit relationship, at Z approximately $> 2$ mm$^6$ m$^{-3}$, should be decreased
relative to the actual slope of the PV11 best-fit line. Readers who view PV11's Fig. 11 will
conclude that this interpretation is not warranted.

As is discussed in Sect. 1, computation-based W-band S/Z relationship have inputs from

parameterized descriptions of density, shape, fall speed, PSD, and particle size. The
computation-based S/Z relationships are in the top three rows of Table 1; the subsequent two
rows of Table 1 have S/Z relationships that resulted from a hybridization of measurements and
calculations (PV11 and Falconi et al. 2018).

We now compare our snowfall rates (fourth column of Table 6) to snowfall rates where

they plot on an S/Z relationship line evaluated at one of our attenuation-corrected reflectivities.
The departure between these is reported as a relative S difference expressed as $|(S_{HP}-S)|/S$ where
$S_{HP}$ is from Table 6 and where S is on an S/Z relationship line. All possible comparisons are
presented graphically in Fig. 12. Table 1 has both the minimum relative S differences and the
salient maximum relative S differences. The comparisons will be discussed in the order of
presentation in Table 1.
In comparisons of our snowfall rates and the upper-limit S/Z relationship line from Hiley
et al. (2011) the relative difference is no smaller than 0.7 and 1.0 on 15 December and 3 January,
respectively. These minimum relative differences exceed the hotplate precision (Sect. 2.4) by at
least a factor of two. It is concluded that our paired values of undercatch-corrected precipitation
rate and attenuation-corrected radar reflectivity provide evidence that a calculation of S based on
the Hiley et al. (2011) upper-limit, when applied to rimed snow particles, is associated with a
low-biased estimate of S. A retrieval based on Hiley et al.'s average S/Z relationship (not
shown), which bisects the orange region in Fig. 12, corresponds to an even larger low bias. This
is a concern because Hiley et al. (2011) used their average S/Z relationship to retrieve global
snowfall distributions and since global observations reported in Wang et al. (2013) document the
frequent occurrence of supercooled liquid within snowing clouds.
Figure 12 shows the separation between our measurements and the Matrosov (2007)
calculation. The separation is about a factor of two (minimum relative difference = 1.4) for the
points obtained on 15 December 2016 and corresponds to an underestimation of S (low bias)
when compared to our measurements. The points from 3 January 2017 plot at an attenuation-
corrected reflectivity smaller than the lower-limit of the calculation (Matrosov 2007). Since the
particle images (Fig. 11a-b) reveal no evidence of the particle type modeled by Matrosov (2007)
(aggregates), it is not surprising that the Matrosov S/Z relationship is not representative of our
measurements.
One plausible reason for the low bias discussed in the previous two paragraphs is the
smaller density implicit in most computationally-based S/Z relationships and especially those
which assume that snow particles are crystals. Densities are quite different for crystals versus
that for rimed snow particles. For example, in Brown and Francis (1995), assuming a 2 mm
crystal, the density is ~ 30 kg m$^{-3}$, whereas in PV11 (their Eq. 1), assuming a 2 mm graupel
particle, the density is ~ 200 kg m$^{-3}$. Because aggregates are collections of crystals, this
comparison of crystal and graupel densities also seems relevant to a comparison of graupel and
aggregate snow particle densities.
Figure 12 compares our $S_{HP}/Z'$ data pairs to the SSKB S/Z relationship line and Table 1
presents the relative differences between the data pairs and the SSKB line. Compared to the S/Z
relationship represented by the top of the orange region in Fig. 12, and compared to the Matrosov
2007 relationship, the SSKB line plots closer to our data points (minimum relative difference ~
0.3). We note that the only instances of $S_{HP} < S$ are three of four comparisons of our
measurements to the SSKB relationship. A possible reason for this is that the density applied in
SSKB (Table 1) is not entirely representative of conditions during our study. An analysis of the
sensitivity of the SSKB to a change in density is needed to investigate our assertion.
Comparisons of our $S_{HP}/Z'$ data pairs and PV11's best-fit line are also in Table 1. The
table demonstrates that the agreement is reasonable - minimum relative difference no larger than
0.3 – and Fig. 12 shows that our data pairs plot at or above the PV11 best fit line.
Based on data from PV11 and our $S_{HP}/Z'$ data pairs, as well as the S/Z relationship
abbreviated SSKB, it is expected that the S/Z relationships reported by Falconi et al. (2018) for
rimed snow particles (Sect. 1) would plot higher in S-versus-Z space than is illustrated in Fig. 12.
Notably, only the upper-end of the Falconi et al. lines (i.e., at $Z > 8$ mm$^6$ m$^{-3}$) plot above the
upper-limit that Hiley et al. (2011) developed for unrimed snow particles. A plausible
explanation for the lower-than-expected S/Z relationships of Falconi et al. is now offered.
Falconi et al. used liquid water path as a proxy for the extent of snow particle riming (von Lerber
et al. 2017). A consequence may have been that the proxy did not dependably exclude unrimed
snow particles (crystals and aggregates) from the riming categories of Falconi et al. If this was
the case, then the data groupings that were the basis for the Falconi et al. S/Z relationships may
have been affected. When applying the heavily-rimed S/Z relationship of Falconi et al. with our
$S_{HP}/Z'$ data pairs we find that the minimum relative differences are 0.6 (December 15) and 8.5
(January 3) (Table 1). Additionally, the differences are 0.5 (December 15) and 5.9 (January 3)
when applying the moderately-rimed S/Z relationship of Falconi et al. (results not shown).
Further research is needed to resolve the reason for the mismatch between the snowfall
rate/reflectivity pairs reported here and the S/Z relationships reported in Falconi et al.
Our conclusion that the upper-limit S/Z relationship from Hiley et al. (2011)
underestimates S would be modified if our WCR-derived reflectivities were negatively biased.
Assuming the reflectivities are negatively biased by 2.5 dBZ, the minimum relative differences
discussed previously are no smaller than 0.1 and 0.3 on 15 December and 3 January,
respectively. A bias in reflectivity of this magnitude cannot be ruled out but neither can a
positive bias of the same magnitude (Sect. 2.3). The latter increases the minimum relative
differences to 1.6 and 2.2 on 15 December and 3 January, respectively. In each of these
calculations we have summed the attenuations (Table 4) with ± 2.5 dBZ and used Eq. 1 to
calculate error-perturbed reflectivities.
The scatter of measurements in Fig. 12, the plausibility of a -2.5 to +2.5 dBZ bias in
WCR reflectivity measurements, and error in measurement of S (Sect. 2.4), indicate that refined
techniques will be needed in future investigations which apply the approach described here.
Taking into consideration the goal of evaluating snowfall rates from space, some advance in
satellite remote sensing also seems warranted. One issue is diagnosing where riming is occurring
within clouds. Both lidars and radiometers can sense supercooled liquid water from space (e.g.,
Battaglia and Panegrossi, 2020), and if combined with Doppler radars operating at multiple
wavelengths, can diagnose precipitation attributable to rimed snow particles. Despite limitations
of the multiple-wavelength Doppler method, for example in scenarios with vertical air speed
comparable to and larger than particle fall speed (Vogl et al. 2022), the method has been
validated in ground-based field studies (Kneifel et al. 2015; Mason et al. 2018). Technical
challenges also remain for implementing the method from space (Battaglia et al. 2020).
**5 – Conclusions**
We have reported surface measurements of S and near-surface measurements of Z. The
latter came from overflights of a ground site, where a precipitation gauge was operated, and were
acquired using an airborne W-band radar. The values of Z were corrected for attenuation.
The reported $S_{HP}/Z'$ pairs plot at or above the S-versus-Z best-fit line of PV11 (Fig. 12)
and the minimum relative S difference (Table 1) is no larger than 0.3. The PV11 data came from
airborne measurements of W-band reflectivity, acquired within ± 100 m of flight level, and from
coincident measurements of snow particle imagery. PV11 used a density-size function and a fall
speed-size function, and measurements (PSD and particle images) to calculate S for snow
particles that were classified as both rimed crystals and graupel. This classification is also
consistent with the particle imagery we have presented (Fig. 11).
We have documented a substantial difference in comparisons between our snowfall rates
and reflectivity-dependent S values calculated using an upper-limit S/Z relationship for unrimed
snow particles (Hiley et al. 2011). Here the minimum relative S differences are 0.7 and 1.0 for
our two overflights and in a comparison to our measurements correspond to an underestimation
of snowfall rate (Fig. 12). The relative differences are approximately a factor of two larger than
the precision of our snowfall rate measurement. We also report a substantial difference, and S
underestimation compared to our measurements (Fig. 12), for the comparison made to an S/Z
relationship which assumes the snow particles are aggregates (Matrosov 2007). The snowfall rate
underestimates obtained using both Hiley et al.'s and Matrosov's S/Z relationships (Fig. 12) are
perhaps expected given that the density factored into those S/Z calculations is small compared to
that for rimed snow particles. It is also expected that the larger density and spherical shape
applied in the SSKB S/Z relationship contributed to the better agreement (minimum relative
difference $\sim 0.3$) with our $S_{HP}/Z'$ pairs. Our conclusion is that some snowfall retrievals (e.g.,
Hiley et al. 2011) will underestimate S for weather targets containing rimed snow particles. We
also state that our conclusion is at odds with measurements and analysis in Falconi et al. Those
researchers reported S/Z relationships for rimed snow particles which in instances with $Z < 8$
$mm^6$ $m^{-3}$ plot below the upper-limit of Hiley et al. (Fig. 12). The consequence is that the
minimum relative S difference in our comparison to Falconi et al. (assuming Falconi et al.'s
heavily-rimed classification) is comparable to and larger than in our comparison to the Hiley et
al.'s upper-limit S/Z relationship.

New research is needed to refine the S/Z relationship for rimed snow particles. This could

be computational – e.g., investigation of the utility of parameterizing S in terms of both Z and
density – or could be observational. Unlike the investigation of PV11, where only an airborne
platform was employed, we have demonstrated that useful information can be obtained using
coordinated ground-based and airborne systems. Another approach would be with only ground-
based instrumentation. This would avoid some of the complications encountered in this study,
including  W-band attenuation and a reliance on particle imagery acquired aloft. A study with
both ground-based and airborne systems would also be useful for understanding an S/Z
mismatch apparent at $Z < 8$ mm$^6$ m$^{-3}$. Elements of the mismatch are the measurements reported
here, PV11's best-fit line, and the measurement-based S/Z relationships reported by Falconi et al.
(2018). These three research teams reported measurements relevant to the development of an S/Z
relationship for rimed snow particles.
**6 – Appendix**
This appendix explains how HP (hotplate) and WCR (Wyoming Cloud Radar) averages
were evaluated. The scheme starts with an HP averaging interval (duration 60 s) and derives a
WCR averaging interval and a WCR averaging domain. The latter encompasses a subset of the
altitude-time crossection sampled by the WCR. The top boundary of the domain was derived
using vertical-component Doppler velocities within the interval/domain. Because of this
dependence, the line defining the top boundary was derived iteratively.
With the overflight time symbolized $t_0$, the beginning and ending times of two 60-second
HP averaging intervals are
$$t_{HP,B} = t_O \tag{A1}$$
$$t_{HP,E} = t_O + 60 \tag{A2}$$
Since two adjacent HP averaging intervals are evaluated in this analysis, we express the
averaging times with the following recursive equations
$$t_{HP,B}(i) = t_O + i \cdot 60 \tag{A3}$$
and
$$t_{HP,E}(i) = t_O + (i+1) \cdot 60. \tag{A4}$$
In Eqs. A3-A4 the index is $i \in \{0, 1\}$. A special case with $i = 2$ is also analyzed (Sect. 3.5).
Analogous to the recursion in Eq. A4, the ending time of a WCR averaging interval is
$$t_{WCR,E}(i) = t_O - i \cdot 60 \cdot v_w / gs .$$    (A5)
Here $v_w$ is a wind advection speed (discussed below) and the second term on the rhs is a wind
advection distance divided by the WKA (Wyoming King Air) ground speed ($gs$). Analogous to
the Eq. A5, the beginning time of a WCR averaging interval is
$$t_{WCR,B}(i) = t_{WCR,E} - (i+1) \cdot 60 \cdot v_w / gs$$    (A6)
The wind advection speed ($v_w$) in Eqs. A5-A6 was calculated using an altitude-
dependent west-to-east wind velocity ($u$) and an altitude-dependent south-to-north wind velocity
($v$). These altitude-dependent component velocities were calculated using the horizontal wind
vectors in the penultimate and last columns of Table 3. Plots of the component velocities versus
altitude and the linear functions used to relate component velocities to altitude are presented in
Figs. A1a-b.
An altitude ($z' = 3400$ m) was assumed for evaluating the horizontal wind advection
vector. This is the altitude of the ridges west and northwest of the HP site (Figs. 3a-b).
The WKA track vector (Table 3) defines the vertical plane of the WCR measurements.
We assumed that wind advection of snow particles occurred parallel to this vector. With the
assumption stated in the previous paragraph, the horizontal wind advection speed ($v_w$) was
calculated as the projection of the horizontal wind vector onto the track vector.
$$v_w = \frac{u(z') \cdot gs_x + v(z') \cdot gs_y}{\left(gs_x^2 + gs_y^2\right)^{1/2}}$$    (A7)
In Eq. A7 the west-to-east and south-to-north components of the track vector are symbolized $gs_x$
and $gs_y$. Vector representations of the track vector are in Table 3. On 14/15 December 2016 and
3 January 2017, the values of $v_w$ are 7.4 and 8.9 m s$^{-1}$, respectively.

In addition to the properties $gs$ and $v_w$ used to evaluate Eqs. A5-A6, a WCR averaging

interval/domain was evaluated using a snow particle downward speed (Eq. A8).

$$v_p = |<V_D>| + \sigma_{V_D} \qquad\qquad (A8)$$

Here, $<V_D>$ is the average of Doppler velocities within an averaging interval/domain,
$|<V_D>|$ is the absolute value of the average, and $\sigma_{V_D}$ is the standard deviation of the average.
On both the lhs and rhs of Eq. A8, all terms are greater than zero.

We interpret $v_p$ as the maximum likely snow particle speed toward the ground. There are

three reasons for this: 1) For the WCR averaging intervals/domains we analyzed, values of
$<V_D>$ were consistently less than zero (Table 6). This indicates that snow particles (on
average) were moving toward the ground. 2) Again, for the WCR averaging intervals/domains
we analyzed, $\sigma_{V_D}$ was comparable to $|<V_D>|$. This indicates that turbulent eddies transported
snow particles upward and downward at a speed comparable to their downward speed in still air.
3) The $V_D$ are reflectivity weighted (Haimov and Rodi 2013) and are thus indicative of the
motion of the largest particles within an averaging interval/domain.

We now focus on the top boundary of a WCR averaging interval/domain. Figures 6b and

6d have representations of the boundary. The slope defining this boundary was calculated as
$-v_p \cdot gs / v_w$. That is, particles below this boundary moved downward sufficiently fast and
horizontally sufficiently slow to advect reasonably close to the hotplate. Starting with diagnosed
values of $gs$ and $v_w$, the values of $v_p$ and slope, were derived iteratively. The precision of the
derived $v_p$ is $\pm 0.1$ m s$^{-1}$.

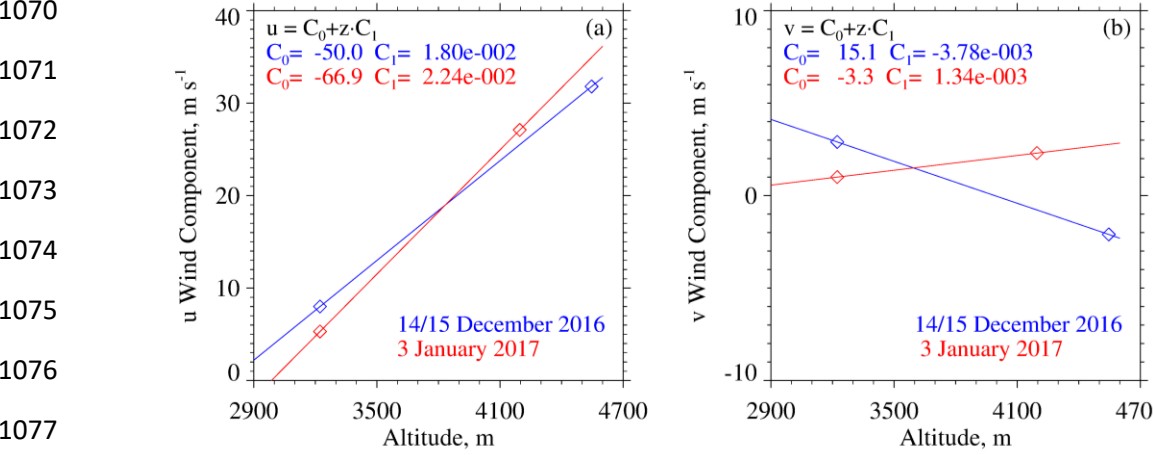

Figure A1 – (a) West-to-east ($u$) wind velocity derived using measurements from the WKA and

the AmeriFlux (AF) tower. Also shown is the linear function used to relate $u$ to altitude. (b)

South-to-north ($v$) wind velocity derived using measurements from the WKA and AF. Also

shown is the linear function used to relate $v$ to altitude. WKA and AF velocities are presented as

vectors in the penultimate and last columns of Table 3.

Data Availability. The WKA and WCR measurements can be obtained from the SNOWIE data
archive of NCAR/EOL, which is sponsored by the National Science Foundation. Hotplate gauge
measurements are at https://doi.org/10.15786/20103146. The US-GLE AmeriFlux measurements
are at https://ameriflux.lbl.gov/. The Brooklyn Lake SNOTEL gauge measurements are at
https://www.wcc.nrcs.usda.gov/snow/. Merged Hotplate, SNOTEL, and AmeriFlux data
sequences from 14/15 December 2016 and 3 January 2017 are in Snider (2023).

Author contributions. JS and MB wrote the grant proposal that funded this research. Field
measurements were performed by SF, SM, SH, MB, and JS. SF wrote her MS dissertation, and
this was adapted for this paper by JS. KS processed the snow particle imagery. AM maintained
the measurement sites. All authors contributed to the editing of this paper.

**Acknowledgements –**
We acknowledge technical assistance provided by David Plummer, Larry Oolman, Zane
Little, Brent Glover, Edward Sigel, Thomas Drew, and Brett Wadsworth. We thank SNOWIE
project PI Jeffery French, who provided the flight data, Gabor Vali who provided the S/Z data
points in Fig. 12, and John Frank and John Korfmacher who acquired  the GLE-US AmeriFlux
data set. This work was supported by the United States National Science Foundation (Award
Number 1850809) and the John P. Ellbogen Foundation.

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
