# Peer review of "W-band S/Z Relationships for Rimed Snow Particles: Observational Evidence from"

_Atmospheric Measurement Techniques, 2022_

## Author Comment (AC1)

Referee1

We appreciate your review and critique of the manuscript. Thank you.

This manuscript describes a study to relate snow fall rate and W-band reflectivity based on two observational events. Overall, the results of this study may add some new incremental knowledge of mm-wavelength radar-based snowfall remote sensing. However, some revisions are needed.

Main comments.

You should, probably add some information about radar calibration. How well is the radar calibrated?

This was added to the revised Sect. 2.3:

"Ground-based calibrations of the WCR's up-looking antenna and correlations between in-flight retrievals acquired using its up-looking and down-looking antennas were used to estimate the absolute accuracy of the WCR-derived values of dBZ. This is ±2.5 dBZ (PV11)."

Did you account for the two-way radar signal attenuation by gases and hydrometers between the aircraft and the radar resolution gate, which was used?

We did in the revised Sect. 3.2.

What are the uncertainties of the hot plate for measuring snowfall rate? Given that sometimes you are getting negative snowfall rates as much as -0.3 mm/h (Fig.8), these uncertainties can be substantial.

The revised Sect. 2.4 includes a description of the hotplate precision. This was based on a comparison between the hotplate and SNOTEL pillow systems (Marlow et al. 2023). The gauge comparison has 57 paired measurements from the HP (hotplate) and SN (SNOTEL pillow) gauges operated at the HP and SN sites in Figs. 1a-b of the revised manuscript. In the revised Sect. 4, we apply the S precision when considering the departure between our S measurements and computation-based values of S. Marlow et al. (2023) was reviewed at AMS/JAMC; we submitted revisions back to the journal two months ago.

As I understand your results are shown only by a couple of points representing mean Z and S values. Why do not you show more detailed information on the S-Z correspondence?

We do not completely understand your question.

Perhaps you are saying this: Why didn't you consider time intervals smaller than 60 s (one minute) for averaging of the hotplate data? If that is correct, then our rationale is in Sect. 3.5:

"We temporally and spatially averaged the values of $Z$ we compared with time-averaged values of S. There are two reasons for this: 1) As discussed in Sect. 3.1, the WCR did not sample

Z exactly over the hotplate, and furthermore, the width of radar beam at 1500 m range - roughly the distance between the aircraft and the ground at the overflight times - is 30 m and thus considerably smaller than the minimum horizontal distance between the aircraft and the HP. 2)

Compared to the WCR, the hotplate is a relatively slow-response measurement system whose output is commonly averaged over one-minute intervals (Z18)."

Or, perhaps you are saying this: Why didn't you average further forward in time (hotplate) and further backwards in time (WCR)? We addressed this in the revision, Sect. 3.5:

"As discussed earlier in this section, the averaging scheme initializes with 60-second blocks of HP data between $t_O$ and $t_O + 120$ s. When we applied the scheme to data from 3

January 2017, but outside the specified time range, an inconsistency was documented. This is apparent in Fig. 8, where the $t_O + 120$ s to $t_O + 180$ s interval (i.e., the $i = 2$ interval) has negligible average S, while in Fig. 10, the $i = 2$ interval has a non-negligible average Z ($\sim 0.3$ mm$^6$ m$^{-3}$). A

firm explanation is not available for the inconsistency, but a factor may be the convective nature of the fields in Figs. 10a-b. Because of the inconsistency, only averages corresponding to the

$i = 0$ and $i = 1$ intervals were analyzed further."

Note that the Matrosov (2007) relation was derived for $Z > 0$ dBZ. It needs to be stated in the paper and shown in Fig. 12 (like it is done in the PV11 paper).

Yes. In fact, some of Matrosov's points (his Fig. 5b) plot slightly smaller below 0 dBZ. Also, some of his low-Z points are for dendritic crystals while most points in his figure are for aggregates. In the revised Sect. 4, we discussed the relevance of Matrosov's calculations as a comparator for our measurements:

"Figure 12 shows our S/Z measurements after we corrected the reflectivities for attenuation. Below we compare those plotted S/Z pairs to calculations reported Hiley et al.

(2011), but first, we consider the computational S/Z relationship reported by Matrosov (2007)

and its relevance to our measurements. Since the particle images (Figs. 11a-b) reveal no compelling evidence for the aggregates modeled by Matrosov (2007), a model based on that particle type is not a useful comparator. Moreover, the overlap of PV11's S/Z measurements and

Matrosov's S/Z calculations has already been discussed in the literature (PV11). However, before going forward, two clarifications will be made about PV11's data points in Fig. 12: 1)

Presentation clarity was what guided our selection of the S and Z axis ranges in this figure but with the consequence that 32 of PV11's S/Z pairs are not shown at $Z > 10$ mm$^6$ m$^{-3}$. 2) The scatter of PV11 data at the largest values of Z in Fig. 12, combined with the fact that PV11

points at $Z > 10$ mm$^6$ m$^{-3}$ are not shown, could lead to the interpretation that the slope describing the relationship at Z approximately $> 2$ mm$^6$ m$^{-3}$ should be decreased relative to the slope of the

PV11 best-fit line. Readers who view PV11's Fig. 11 will conclude that this interpretation is not warranted."

How the reflectivities were averaged? Did you average them in linear scale (mm^6/m^-3) or in the logarithmic scale (i.e., in dBZ units)?

In the original submission, and in the revision, we averaged the Z values (mm$^6$ m$^{-3}$).  In the revision (Sect. 2.3), we explicitly state that.

How well the snowfall rate and reflectivity measurements were collocated? What was the vertical separation between radar Z and hotplate S measurements used in analysis of Z -S pairs?

Section 3.6 explains this:

"Figure 11a shows imagery from 12 s of measurements acquired near the end of the sequence in

Fig. 9a (00:01:02 to 00:01:14). This time interval was selected by tracing forward from $t_O$, along the slope of the fall streaks, to the flight level."

From Fig. 9a you can see the vertical separation between flight level and the altitude of the hotplate.  The hotplate is at the overflight time (~3010 m) and the flight level is at ~4550 m.  The vertical separation is therefore 1540 m. That vertical separation is also equal to the pathlengths for vapor and snow particles in Table 3 (revised manuscript) where attenuation is estimated.

Section 2.3: How did you separate components of the Doppler velocities (i.e., the reflectivity- weighted fall speeds and vertical air motions)?

We did not do that. Rather, we averaged Doppler velocities in a WCR averaging interval/domain and used Eq. A8 to calculate $v_p$ . The latter is our "maximum likely snow particle speed toward the ground. Details are in the revised Sect. 3.5 and in the revised

Appendix.

Was your assertion that particles were rimed based for the most part only on the analysis of the

2DP particle images?

We used both optical array probes. This is stated in Sect. 3.6.

Did you utilize 2DS particle measurements?

Yes. This is stated in Sect. 3.6.

You suggest that the 2DP particle images are representative of those that fell from the flight level toward the hotplate. It might be not so since the height separation was very significant.

Yes, but we don't have ground measurements of particle shapes, so, Sect. 3.6 and Figs. 11a-b are the best we can do.

Minor comments

Line 91: what are rho_ 1 and rho_3 ?

We thought this was clear from Sect. 1. Since it wasn't, we added the following to the revised

Sect. 3.7:

"…In the figure legend, results from PV11 are specified as $S(\rho_1)/Z$ because those authors applied the lower of two density-size functions ($\rho_1$) with airborne measurements of optical particle images to calculate the snowfall rates (Sect. 1). Our data pairs plot above the $S(\rho_1)/Z$ line but within the variability of PV11's measurements."

The manuscript could benefit from additional editing.

Yes. We worked on that.

I wonder if you need any permission to reproduce the figure from PV11 paper (their Fig. 11), which is copyrighted by the AMS.

We don't know. In the Acknowledgements, we do acknowledge Gabor Vali for providing data values published in Fig. 11 of PV11.

---

## Author Comment (AC2)

Referee2

We appreciate your review and critique of the manuscript. Thank you.

The manuscript presents a field experiment in which airborne W-band reflectivity is matched with ground measurements of snowfall rate to investigate the Z-S relationship for rimed particles. The topic is very important for the precipitation community because the uncertainties in the microphysics still lead to very big uncertainties in the precipitation retrievals. The authors follow up from a series of previous papers, but in particular from the Pokharel and Vali 2011 (PV11) in which a full range of particle types is assumed and the precipitation rate is calculated from particle density assumptions. In this manuscript the authors focus on a specific particle type, rimed particles, for which precipitation rate is usually underestimated using "conventional" Z-S relationships.

Despite the great importance of the topic, the manuscript doesn't really provide a Z-S relationship for rimed particles as the title would suggest. Most of the manuscript is focused on the description of the methodology used to calculate the relationship, and very little space is dedicated to actual results. 4 points are really not enough to derive a Z-S relationship and the conclusions just state that the measurements of this field campaign fit within PV11 variability. The fact that rimed particles were not really well represented by published Z-S relationships was already known so the fact that this manuscript does not present a new Z-S relationship specific for rimed particles doesn't match with what the title suggests.

The title was revised, and the abstract was revised. Readers of the abstract will see that the number of S/Z pairs in our analysis is smaller than in PV11.

In the revision, we distinguish our work against the studies of PV11. We made direct measurements of S while PV11 derived S using particle imagery. We think this makes our contribution significant, despite the smaller number of points.

Probably the use of a ground based W-band pointing radar would have helped with the availability of Z-S points, aided by the aircraft overpass to confirm the presence of riming with the cloud probes.

We agree. At the end of the revised Sect. 5, we state the following:

"New research can also refine the S/Z relationship for rimed snow particles. This could be computational – exploring the utility of parameterizing S in terms of both Z and density – or could be observational. Unlike the investigation of PV11, where only an airborne platform was employed, we have demonstrated how useful information can be obtained with ground-based and airborne systems. Another approach would be with collocated ground-based instrumentation, for density and particle imaging, and for measuring wind, snowfall rate, and radar reflectivity. This would avoid some of the complications encountered in this study, including W-band attenuation and a reliance on particle imagery acquired aloft. A close-range measuring radar might also allow retrievals closer to the surface than in this work. Improvement of methods that remotely sense supercooled cloud water are also needed."

Given the availability of data (I assume no more aircraft overpasses are available at the site, otherwise they would have been used),…

The two flights analyzed were two of three test flights flown from Laramie in preparation for the

SNOWIE campaign (Tessendorf et al. 2019). The other test flight did not fly over the ground site.

I suggest to stress more the position of the Z-S points in fig. 12, trying to figure out what differentiates these 4 points from all the other points under the black best fit line or from the

Matrosov 2011 range.

Following your critique, and that of Referee3, who brought Hiley et al. (2011) to our attention, we revised this section.  In the revised text, we compare our measurements to Matrosov's (2007)

calculation, as in the original submission, and we also compare our measurements to Hiley et al.

(2011).

Attached here is revised text, from Sect. 3.7, relevant to your criticism:

        "Our S/Z pairs are presented in Table 5 where the indexes ($i = 0$ and $i = 1$) are used to indicate results derived for the averaging intervals. Here, the reflectivities are not corrected for attenuation, however, in Fig. 12, the attenuation-corrected reflectivities are plotted. Uncorrectedreflectivities from Table 5, attenuations from Table 3, and Eq. 1 were used to calculate the corrected reflectivities…."

[Figure]

"Figure 12 – Snowfall rate versus radar reflectivity. Colored circles indicate attenuation-corrected reflectivities (Table 3, Table 5, and Eq. 1) for the $i = 0$ and $i = 1$ averaging intervals. The $S(\rho_1)/Z$ points are a subset from PV11's Fig. 11 ($0.01 < Z < 10$ mm$^6$ mm$^{-3}$). Also plotted is the PV11 best-fit line (black), the S/Z relationship from Matrosov (2007), the S/Z relationship abbreviated SSKB (Sect. 1), and the swath of S/Z relationships, for crystals, from Hiley et al. (2011)."

Here, from the revised Sect. 4, is discussion of Fig. 12. This is also relevant to your criticism.

   "We now evaluate departures between our S measurements and S/Z calculations from Hiley et al. (2011). Each of the departures will be evaluated as the vertical distance between the top of the orange region in Fig. 12 and our S/Z data points. Reflectivities at the top of the orange region were calculated using attenuation-corrected reflectivities (Eq. 1) and the upper-limit S/Z equation from Hiley et al. (2011) ($S = 0.21 \cdot \left( Z' \right)^{0.77}$; Sect. 1 and Eq. 1). In terms of a relative difference, expressed as ($S_{HP}$-S)/S and with $S_{HP}$ an attenuation-corrected snowfall rate, the departures are no smaller than 0.9 and 1.1 on 15 December and 3 January, respectively. These minimum relative differences exceed the hotplate precision (Sect. 2.4) by approximately a factor of three. We therefore conclude that our paired values of surface-measured precipitation and aircraft-measured radar reflectivity, after correcting for attenuation, provide evidence that a calculation of S based the Hiley et al. (2011) upper-limit, when applied to rimed snow particles, is associated with a low-biased estimate of S.”

On the other hand, I understand that this journal is about atmospheric measurement techniques, so if the goal is to describe the methodology to match aircraft with ground based observations, that is not really clear from the title and the abstract. As I said earlier, my expectation here is to find a new Z-S relationship for rimed particles. Based on what you decide the goal of the manuscript is, please revise accordingly.

In addition to modifying the title and abstract, we addressed this by adding goals to the revised

Sect. 1.

“The goals of this paper are as follows: 1) to describe measurements of undercatch- corrected liquid-equivalent snowfall rate (S, mm h$^{-1}$) that were paired with W-band measurements of reflectivity (Z, mm$^6$ m$^{-3}$) ; 2) to contrast the measurement-based S/Z pairs against calculated S/Z relationships commonly applied in retrievals of S based on reflectivity; and 3) to investigate why the acquired data set deviates from predictions of some calculated S/Z

relationships.”

Also as a general comment, there are too many not needed figures in this manuscript, I provided some suggestions to consolidate them.

Figures 7a and 8a are removed from the revised manuscript.

Specific comments:

Section 2.1 and in general when you mention AF environmental data. It is not clear to me when you actually use this dataset in your analysis since HP already has the data needed to calculate precipitation rate. Probably I missed it, but I would suggest to be more clear so it could be more obvious.

This is clarified in the revision. The AF data was used to derive the following: Absolute humidity (Sect. 3.2), cloud base altitude (Sect. 3.2), horizontal wind advection speed (Sect. 3.5), and adiabatic cloud liquid water path (Sect. 3.7). We used AF measurements for these properties because the hotplate T measurement is known to be high biased during daytime (Marlow et al.

2023). Marlow et al. (2023) was reviewed at AMS/JAMC; we submitted revisions back to the journal two months ago.

But on the other side, how far are the two sites? we know environmental conditions change a lot, especially in mountain environment, could the conditions be very different in this case?

AF and HP were separated horizontally by 2000 m and vertically by 190 m. SN and HP were separated horizontally by 1200 m and vertically by 110 m. Site altitudes are in Fig. 1a.

Is it actually reliable to use that data as it was at HP? And the same is for the SNOTEL site,
would it actually reflect the HP situation?

The AF thermodynamic measurements (T/RH/P) were acquired on a tower at a long-term climate monitoring site (AmeriFlux). The exact altitude of that measurement is in the footnotes of Table 2. Relevant to your question, here is what we know about the ground sites: 1) The vertical separation of AF and HP, and 2) that the winter-season wind flow is nearly always directed approximately from AF to HP. From those characteristics, and the dry adiabatic temperature lapse rate, we expect the temperature difference AF - HP to be no smaller than -2 K.

If you look at the sequences from HP and AF (Data Availability Statement; https://doi.org/10.15786/20247870), you will see that the AF - HP temperature difference, at night (see above discussion of the HP's daytime temperature measurement bias), conforms to our expectation. Hence, we think it is reasonable to assume the AF thermodynamic measurements are representative of the region surrounding the three ground sites (AF/SN/HP). This region is shown in Figs. 3a-b.

The consistency of the SN and HP snowfall measurements is discussed in Sect. 2.4 (revised manuscript) and in Marlow et al. (2023).

Regarding the AF-derived horizontal wind velocity, we do not have a check on how representative that is for the AF/SN/HP region. We do know that the measurement was made above the tree tops (the anemometer was/is deployed at the top of a tower) and that the measurement system (propeller anemometer) is reliable.

Section 2.4, you describe the hotplate and all the bias corrections needed, included a comparison with a fenced precipitation gauge. Why isn't the HP inside a fence?

We apply an algorithm which assumes the hotplate is _not_ within a fence.  This is discussed in

Sect. 2.4 of the revised manuscript.

Section 3.3, lines 287-291: why mentioning this previous attempt to compare wind speeds if data sets are difficult to interpret and they do not provide useful results for this work?

Because we reported, in a conference presentation, comparisons of hotplate-derived and Vaisala- derived wind speeds.  We later found the problem with the Vaisala-derived speeds.

What is the point to show up- and down-looking reflectivities? Up-ward ones are not needed for this work…

There are three reasons for this. 1) In Sect. 3.6 we discuss the fall streaks at ~ z = 5500 m in Fig.

5a (i.e., above the flight level in the up-looking height-time crossection). 2) People would ask for what's above the flight level if we did not show that information. 3) To compare, on one page, the two weather systems (i.e., one has relatively large reflectivities, is deeper and stratiform, the other has smaller reflectivities, and is shallow and convective).

…actually these plots are a repetition of figures 9 and 10 (except for the up-ward reflectivities).

Vertical winds can be consolidated into figs 9 and 10 too, focusing on the portion of the overpass that is actually of interest for the analysis.

We think we have crafted things effectively and logically. Please consider the revised manuscript. Here is how the presentation evolves from Figs. 5a-d, to particle imagery (Sect. 3.6), to Sect. 3.7 (S/Z Relationships), and to Fig. 12:

What is shown in Figures 5a-d (Sect. 3.5) ends at the overflight time. Figures 6a-d explain the averaging. Figures 7 and 8 show the ground measurements and ground-measurement averaging intervals. Nearly at the end of Sect. 3.5, we introduce Figures 9a-b and 10a-b. These show the

WCR measurements prior to and after aircraft's overflight. We also state why the time axes are different in Figures 9a-b and 10a-b (compared to Figs. 5a-d), and that the WCR "structures" in

Figs. 9a-b and 10a-b will be discussed in the following section (i.e., Sect. 3.6, Snow Particle

Imagery).  Section 3.5 ends with Table 5.  The Table 5 has the averages.  The averages are the basis for Fig. 12, Sect. 3.7 (S/Z Relationships), and Sect. 4 (Results).

Line 433-434, the meaning of the slopes is not really clear if the reader hasn't read the appendix yet. I would suggest to add a sentence explaining why the HP line is flat while the WCR one has a slope (and then refer to appendix for details).

We revised this portion of the manuscript and revised Fig. 6.  Here is the revised text:

"The HP measurements were averaged over two adjacent 60 s intervals. The first extends from $t_O$ to $t_O + 60$ s (Fig. 6a) and the second from $t_O + 60$ s to $t_O + 120$ s (Fig. 6c).  In Fig. 6a and in Fig. 6c, $t_{HP,B}$ symbolizes an interval's beginning time and $t_{HP,E}$ symbolizes an interval's ending time. Formulas describing how these times were related to the beginning and ending times of the corresponding WCR averaging intervals are in the Appendix. Fig. 6b is a schematic of the first WCR averaging interval and Fig. 6d is a schematic of the second. Again, the subscripts "B" and "E" are used to indicate averaging beginning and ending times. Figures 6b and 6d both have lines at the tops of an averaging interval/domain. The slopes of these lines are proportional to the ratio of two speeds. These speeds are a maximum likely snow particle speed toward the ground ($v_p$) and a horizontal wind advection speed ($v_w$). The $v_p$ was calculated using averaged vertical-component Doppler velocities and $v_w$ was calculated using a vertical profile of horizontal winds, based on WKA horizontal wind measurements and AF horizontal wind measurements (Figs. A1a-b), and using the WKA track vector (Table 2). An altitude ($z' = 3400$

m) was assumed in the calculation of $v_w$. This is the altitude of the ridges west and northwest of the HP site (Figs. 3a-b). Picking the altitude to be either $z' = 3200$ m or $z' = 3600$ m does not alter our findings."

Figure 6: I am not sure this figure is needed or can probably be moved to the appendix. I find it a bit confusing.

We revised Fig. 6.

Figure 7b is the same as fig. 2, just extended to reflect the situation around the observation time.

I would try to consolidate the figures.

Figures 7a and 8a (both had wind speed at the hotplate) were eliminated from the revision.

As I mentioned before, despite the presence of fig. 6, the averaging intervals are not clear and confusing. The appendix should be for details, not for the general understanding of what we are looking at. For example the difference between i=0 point being after t0 for HP and before for

WCR should be stated somewhere in the text (not only in the appendix). Or the meaning of the

WCR slope.

Figure 6 was revised.

Minor comments:

In the abstract you refer to 'published Z-S relationship' which sound like a very specific one (I

assume you are referring to PV11). It is probably good to mention it.

Yes, in the revised abstract we did that.

line 309: add 'forced through the origin, RED LINE'.

Yes, in the revised manuscript we did that.

Line 366: provide a time reference for the ridgeline as you did for the last 3 seconds.

Yes, in the revised manuscript we did that.

Figure 5, the plot at the end goes outside the axes (red line).

Yes, in the revised manuscript we fixed that.

Figures 7a and 8a are never mentioned in the text, either mention them or remove.

Yes, in the revised manuscript those two panels are removed.

Figures 9b and 10b, usually doppler velocity has a blue/red colormap, you might consider it for consistency with other publications or just for differentiating it from the reflectivity plot on figs

9a and 10a.

Yes. This was done in the Doppler velocity panels of Figs. 9 and 10.

Line 629: 'within the variability' – maybe in fig. 12 you can plot the PV11 variability to make it more clear.

We did not do that, but Fig. 12 was substantially modified in the revision.

Line 693: in Kulie et al the threshold is 0 dBZ.

That sentence was removed from the revision.

---

## Author Comment (AC3)

Referee3

We appreciate your review and critique of the manuscript. Thank you.

This manuscript advertises observational evidence from combined ground-based snowfall rate (S) and airborne W-band radar reflectivity (Z) measurements that rimed frozen hydrometeors are associated with somewhat unique Z-S relationships.   These types of studies are desperately need to more accurately characterize the sensitivity of W-band reflectivity to different particle microphysical characteristics, so I laud the authors on their attempts to constrain Z-S relationships for rimed situations using observational assets.    My main concern is the lack of data points presented in this analysis - are the results meaningful since the sample size is so small?   I am not sure how to suggest solving this issue other than collecting and analysing more data.    Conversely, I am very cognizant of how difficult it is to match spatiotemporally disparate datasets like airborne radar to point source measurements of precipitation rates at the ground, so I can appreciate how this study might still be valuable to the community by demonstrating the "atmospheric measurement technique" used so it can be replicated and improved in the future. The manuscript could probably be improved greatly if the narrative leaned more heavily into this aspect of the study. Addressing this issue might be as simple as more forcefully advertising how difficult it is to make such measurements combined with how important it is to collect observational Z-S evidence under rimed conditions in both the introduction and conclusions.   I might be able to offer more impactful suggestions in the future when I digest the manuscript again, but I encourage the authors to think about how to creatively make the narrative more impactful.

The revision has improved explanations of the approach we took (Sect. 1); we also improved on descriptions of our method for acquiring S/Z measurements (Sect. 3.5). Discussion of how our S/Z pairs compare to computed S/Z relationships is also revised (Sect. 4 and Sect. 5). In the revised Sect. 5, we added discussion of possible paths for future studies of S/Z relationships. In sum, we think the revised manuscript is improved in terms of how we describe what we did, how we describe our findings, and in terms of our descriptions of future research needed to better refine S/Z relationships for rimed snow particles.

Specific comments:

Introduction:  I think it's important to note sooner in the introduction that some of the initial S/Z studies performed for W-band radars were purely modeling (i.e., using backscatter calculations from idealised models of frozen ice habits combined with parametrised particle size distributions) studies. This is a very simple way to accentuate the methodological differences (and importance) of observationally-based studies to assess the veracity of idealised modeling studies.

We added a paragraph to the revised Sect. 1. This encapsulates the connections between our observational approach and the computational work of others.

"The goals of this paper are as follows: 1) to describe measurements of undercatchcorrected liquid-equivalent snowfall rate ($S$, mm h$^{-1}$) and how those were paired with W-band measurements of reflectivity ($Z$, mm$^6$ m$^{-3}$) ; 2) to contrast the measurement-based S/Z pairs against calculated S/Z relationships commonly applied in retrievals of S based on reflectivity; and 3) to investigate why the acquired data set deviates from predictions of some calculated S/Z

relationships."

Two further studies of interest (and there are likely more) are Hiley et al. (2011) and Kneifel et al. (2015). Both highlight W-band radar applications for snowfall estimation and also provide analyses that either hint at or explicitly demonstrate how the existence of supercooled water and associated riming complicate Z-S relationships.

When writing the original submission, we were not aware Hiley et al. (2011). The latter is now one of the computational studies we compare to in the revision. Kneifel et al. (2015) is also included in the revision.

Battaglia and Delanoe (2013) and Battaglia and Panegrossi (2020) also demonstrate the global occurrence of snowfall events with supercooled liquid water and Z-S implications.   These studies might provide additional context to frame this study's importance, including W-band attenuation.

The second of these is referenced (revised Sect. 5) because it synergizes lidar, radiometer, and active W-band remote sensing with a views toward retrieving the spatial distribution of supercooled liquid and diagnosing where riming is occurring. Also, the paper's discussion of attenuation helped us in formulating our assessment of attenuation.

I am not very familiar with the hotplate and its history of accurate snowfall rate measurements. While the authors provide some background on previous studies that have been published using hotplates, mostly related to various hotplate precipitation estimates due to various issues (e.g., catch efficiencies, wind speed measurement height, etc.), I still do not see any evidence that this instrument is effective at accurately measuring snowfall rates under various environmental conditions.  I would greatly appreciate at least a few more sentences that describe hotplate performance based on previous studies, including uncertainty estimates.  No snowfall rate measurement device is perfect, but it would nice to see more details regarding the hotplate since this instrument is such an important component of this study.

The revised Sect. 2.4 includes a description of the measurement precision. This was based on a comparison between the hotplate and the SNOTEL pillow systems (Marlow et al. 2023). The gauge comparison has 57 paired measurements from the HP (hotplate) and SN (SNOTEL pillow) gauges operated at the HP and SN locations in Figs. 1a-b. In the revised Sect. 4, we apply the S precision in a discussion of the departure between our measurements and the computational S/Z relationships. Marlow et al. (2023) was reviewed at AMS/JAMC; we submitted revisions back to the journal two months ago.

Somewhat related to the last point, can the authors further quantify (or at least qualitatively describe) the uncertainties related to their spatiotemporal averaging methodology for both airborne radar and ground-based snowfall rate measurements?  What is the sensitivity of the results for slight changes in averaging methodology?

There is discussion of this in the revised manuscript.  The following is from Sect. 3.5.

"The HP measurements were averaged over two adjacent 60 s intervals. The first extends from $t_O$ to $t_O + 60$ s (Fig. 6a) and the second from $t_O + 60$ s to $t_O + 120$ s (Fig. 6c).  In Fig. 6a and in Fig. 6c, $t_{HP,B}$ symbolizes an interval's beginning time and $t_{HP,E}$ symbolizes an interval's ending time. Formulas describing how these times were related to the beginning and ending time of a corresponding WCR averaging interval are in the Appendix. Fig. 6b is a schematic of the first WCR averaging interval and Fig. 6d is a schematic of the second. Again, the subscripts "B"

and "E" are used to indicate averaging beginning and ending times. Figures 6b and 6d both have lines at the top of an averaging interval/domain. The slopes of these lines are proportional to the ratio of two speeds. These speeds are a maximum likely snow particle speed toward the ground (

$v_p$ ) and a horizontal wind advection speed ( $v_w$ ). The $v_p$ was calculated using averaged vertical- component Doppler velocities and $v_w$ was calculated using a vertical profile of horizontal winds, based on WKA horizontal wind measurements and AF horizontal wind measurements (Figs.

A1a-b), and using the WKA track vector (Table 2). An altitude ( $z' = 3400$ m) was assumed in the calculation of $v_w$. This is the altitude of the ridges west and northwest of the HP site (Figs.

3a-b). Picking the altitude to be either $z' = 3200$ m or $z' = 3600$ m does not alter our findings."

The radar blind zone, and what happens within that layer, is incredibly important. The 200 m
WCR blind zone is mentioned in this study in a few locations, but I think the authors need to
mention more prominently that a tacit assumption used in this study (similar to a host of other
airborne or spaceborne radar studies) is that microphysical evolution within the blind zone could
be a major source of uncertainty.   I do not recall any studies that conclusively document how
rimed particle density evolves in the lowest few hundred meters of the atmosphere – presumably
not much – but this is an important to note within this manuscript.   It at least warrants a topic
that should be studied in the future in the conclusion or discussion sections.   It would have been
nice to have additional microphysical measurements at the surface to assess the microphysical
evolution, but I completely understand how difficult it is to procure instrument suites for
fieldwork.
We agree. There is the 200 m deep radar blind zone that encompasses the flight track and the
blind zone immediately above the terrain. The latter is a consequence of ground clutter, and in
our opinion, is more important for our analysis.  Given this, we wrote this in the revised Sect. 5:
"New research can also refine the S/Z relationship for rimed snow particles. This could
be computational – exploring the utility of parameterizing S in terms of both Z and density – or
could be observational. Unlike the investigation of PV11, where only an airborne platform was employed, we have demonstrated how useful information can be obtained with ground-based and airborne systems. Another approach would be with collocated ground-based instrumentation, for density and particle imaging, and for measuring wind, snowfall rate, and radar reflectivity. This would avoid some of the complications encountered in this study, including W-band attenuation and a reliance on particle imagery acquired aloft. A close-range measuring radar might also allow retrievals closer to the surface than in this work. Improvement of methods that remotely sense supercooled cloud water are also needed."

I will likely add further comments later in the review cycle.   But I would like to see the above comments addressed by the authors before I devote more time to more specific comments.

I think this manuscript has potential and could be publishable.   But I encourage the authors to fine tune it further to make it more impactful.

---

## Author Response (AR2)

Referee1

Reconsidered after major revisions.

Were a revised manuscript to be sent for another round of reviews:

I would not be willing to review the revised manuscript.

We appreciate your review and critique of the manuscript. Thank you.

1. It appears that one of the main outcomes of the paper is adding 4 more data points (2 red and 2 blue dots in Fig. 12) to those already suggested in PV11. While it is informative but it is, probably, just an incremental result.

Below are the Conclusions from revision2-Sect. 5.  In this review, you also recommended that results from Falconi et al. (2018) be considered. We did that in this revision (revision2). Discussion of Falconi et al. is in revision2-Sect.1, revision2-Sect.4, and in revision2-Sect. 5.

**5 - Conclusions**

The reported measurements consist of surface measurements of S and near-surface measurements of Z. The latter came from overflights of a ground site, where a precipitation gauge was operated, and were acquired using an airborne W-band radar. The values of Z were corrected for attenuation. The reported S/Z pairs plot at or above the S-versus-Z best-fit line of PV11. However, the points do not depart beyond the variability evident in a replotting of S/Z pairs from PV11. The PV11 data came from airborne measurements of W-band reflectivity, acquired within ± 100 m of flight level, and from coincident measurements of snow particle imagery. PV11 used a density-size function and a fall speed-size function, and measurements (PSD and particle images) to calculate S.

There is an offset between the S points, reported here, and reflectivity-dependent S values calculated at an upper-limit S/Z relationship for unrimed snow particles (Hiley et al. 2011). The offset is larger than the precision of the S measurement. This suggests that a measured Z and the Hiley et al. (2011) upper limit will produce an underestimate of precipitation in scenarios dominated by rimed snow particles.

New research is needed to refine the S/Z relationship for rimed snow particles. This could be computational – e.g., investigating the utility of parameterizing S in terms of both Z and density – or could be observational. Unlike the investigation of PV11, where only an airborne platform was employed, we have demonstrated that useful information can be obtained using coordinated ground-based and airborne systems. Another approach would be with only ground-based instrumentation. This would avoid some of the complications encountered in this study, including W-band attenuation and a reliance on particle imagery acquired aloft. A study with both ground-based and airborne systems would be useful for understanding a S/Z mismatch, apparent at $Z < 8$ mm$^6$ m$^{-3}$, and which is larger than the offset summarized in the previous paragraph. Elements of the mismatch are the S/Z measurements reported by PV11, the measurements reported here, and the measurement-based S/Z relationships reported by Falconi et al. (2018). These three research teams reported measurements relevant to the development of a S/Z relationship for rimed snow particles.

2. I am still concerned about an assumption that microphysical and thermodynamic information inferred aboard the aircraft is representative for the entire layer below, which is more than 1 km thick. Indeed, Figs. 9 and 10 show that there was significant inhomogeneity in the precipitating cloud between the flight level and the ground.

We interpret this as a critique of our analysis of attenuation (Sect. 3.2). An important element of that analysis is the depth of liquid cloud vs. the depth of precipitating snow. We assert (revision2-Sect. 3.2) that the vertical extent of liquid is smaller than the vertical extent of snow particles. This follows because the relative humidity measurements (Table 2) indicated subsaturation ($RH < 100\%$) at the AmeriFlux site (AF). Our assertion is stated in revision2-Sect.

3.2.

3. I tend to think that the two-way attenuation (Table 3 in the reviewed manuscript) might be underestimated. For example, from Table 1 in Liebe et al. 1989 (i.e., the reference used by the authors) one can see that at W-band one-way liquid water attenuation at LWC=1 g/m3 is around

4.6 dB/km. At LWC=0.08 g/m3 and pathlength 1.19 km (the data for the 3 January 2017 case from Table 3 in the reviewed manuscript) it will amount to about 0.9 dB two- way attenuation only due to liquid water. Table 3 shows 0.82 dB total (including attenuation by water vapor, snow, cloud water).

Four clarifications are needed before we address your comment:

A) For cloud water, the extinction coefficient, per $g/m^3$ LWC, varies inversely with temperature.

B) The temperature we applied in the revision1 was the AmeriFlux temperature. This was changed to the Aircraft temperature in revision2. Since the Aircraft temperature is smaller than the AmeriFlux temperature (Table 2), the change increases the extinction coefficient per $g/m^3$

LWC in revision2-Sect.3.2 relative to that in revision1-Sect. 3.2.

C) The extinction coefficient, per $g/m^3$ LWC, in revision2 is larger than the value (4.6 dB/km per

$g/m^3$) in your comment. What we applied (revision2) is 6.1 dB/km per $g/m^3$ (for 20170103). The latter is equal to the ratio of what we state in footnote d of Table 3 (0.49 dB/km for 20170103)

divided by the LWC (0.08 $g/m^3$ for 20170103). The calculation is based on the formula on p. 191

of Vali and Haimov (2001).  The aircraft-measured temperature (Table 2) was applied in the calculation (revision2).

D) Compared to what you state, the pathlength we applied ("Cloud Water", Table 3) is smaller than the value in your comment (1.19 km, for 20170103). We applied 0.59 km (for 20170103).

We applied a smaller pathlength for cloud water, compared to that for oxygen (1.19 km for

20170103), Vapor (1.19 km for 20170103), and Snow (1.19 km for 20170103). This is consistent with what we state in both revision1-Sect. 3.2 and in revision2-Sect.3.2.

Now we address your comment that the attenuations might be underestimated. The "Overall

Two-way Attenuation" in revision2-Table 3 (for 20170103) is 1.01 dB (for 20170103).

Contributions are 0.07 dB (oxygen), 0.17 dB (Vapor), 0.58 dB (Cloud Water), and 0.18 dB

(Snow). In revision2, compared to revision1, attenuation by cloud water is larger because we applied the WKA temperature (colder than the AF temperature applied in revision1). Also compared to revision1, the snow attenuation is larger because we accounted for the profile of

IWC below the WKA. The revision2-Sect.3.2 has details.

Table 3 immediately follows the revision2-Sect.3.2.

Please also note: The vapor concentration (for 20170103) was entered incorrectly in Table 3 of revision1.  The correct value is $1.3 \times 10^{-3}$ kg/m$^3$. This correction has been implemented in revision2.

Dry air (oxygen) attenuation was not considered.

In revision2-Sect.3.2, we accounted for attenuation by oxygen.

4. There were recent studies of W-band Z-S relations in rimed snow specifically, which the authors did not mention (e.g., https://doi.org/10.5194/amt-11-3059-2018 , see table 4 in this reference). If the reviewed paper is published, it would be useful to show relations from this previous AMT-published study (for example, in this manuscript Fig.12) for comparisons with the

PV11 and these author results and discuss differences.

Thank you for bringing the Falconi et al. relationships to our attention. Three S/Z relations from

Falconi et al. (2018) are discussed in revision2.  There are three ramifications: A) Revision2-

Sect.1 has a description of what Falconi et al. (2018) measured and where they report their findings (their Table 2). However, we did not evaluate the S/Z relations that Falconi et al.

derived using TMM-based calculations of Ze. The latter are in Table 4 of Falconi et al. B) Our

Fig. 12b shows the S/Z relations from Falconi et al. C) We discuss Fig. 12b in revision2-Sect.4

and briefly in revision2-Sect.5.

Why Table 3 is on the last page of the manuscript?

The Table 3 is placed correctly within revision2.

Lines 37-38: Reflectivity factor represents the range corrected backscattered power not just the measured power.

This was corrected.

Can you show error bars for your four points in Fig.12?

Error bars were added to Fig. 12a in revision2.

Referee 2

accepted subject to minor revisions

Were a revised manuscript to be sent for another round of reviews:

I would be willing to review the revised manuscript.

We appreciate your review and critique of the manuscript. Thank you.

The authors have made a good faith effort to address the points raised during my initial review. I
appreciate their efforts.

My main concerns have largely been addressed. The revised manuscript's narrative is greatly
improved by an updated and reconfigured introduction and key changes that have been
implemented within the methods and results sections, including expanded content on
measurement uncertainties, W-band attenuation corrections, and placing the results of this study
more effectively within prior research. These changes elevate the revised manuscript
substantially compared to the original submission.

A few minor comments are included below. I hope the authors consider them before the
manuscript is published.

1. Lines 124-125: Since AMT is a European journal with a large international readership, I
recommend adding more geographical context to the site description. Perhaps everyone knows
where Wyoming is located within the larger United States footprint, but I would at least add
"United States", "Rocky Mountains", "Intermountain West", or other generic wording to better
describe the location.

These descriptors complicate our presentation. We did changed the axis labeling in Fig. 1a.

These are now "North Latitude, $^{\circ}$" and "West Longitude, $^{\circ}$." We expect that this adds the necessary geographical context.

2. I do not know how to most optimally incorporate this other minor comment within the
manuscript, but the study described in the manuscript is another example of just how incredibly
difficult it is to extract meaningful information about snowfall properties using disparate
observational sources. Perhaps this notion could be explicitly added in the introduction or
conclusion (or both). The details presented in the methodology and results sections highlight the painstaking steps required to blend airborne and ground-based measurements to extract a few
meaningful data points.
In revision2-Sect.5, we accept that there are complications in the approach we took. We also say
that a significant discrepancy remains among what PV11 and we report compared to what
Falconi et al. (2018) report.  All these research teams report measurements relevant to the
development of a S/Z relationship for rimed snow particles.
3. Related to the last point, the authors did change the revised manuscript title to "On the S/Z
relationship for rimed snow particles in the W-band". This title is rather generic and does not
fully encapsulate the observational aspect of this study. A more appropriate title should
encapsulate the observational complications of deriving S/Z relationships of rimed particles
using combined airborne and ground-based measurements. A few suggestions to consider:
a. W-band S/Z relationships for rimed snow particles: Observational evidence from combined
airborne and ground-based observations
b. The complex task of deriving W-band S/Z relationships for rimed snow particles using
combined airborne and ground-based observations
c. New observational evidence of W-band S/Z relationships for rimed snow particles using
collocated airborne and ground-based sensors
We adopted your first suggestion.  Thank you.
I strongly recommend adding some sort of "observational" component to the title to highlight
that S/Z relationships are derived solely from observations. Adding an airborne and ground-
based component to the title also inherently advertises that matching these disparate data sources
will be a key component of the manuscript and helps alleviate the notion that a rather limited
dataset (4 data points) is extracted from the observations.
These are all minor points, although I would argue that a more effective title could amplify
interest about this manuscript to the larger community.

Referee 3

accepted subject to minor revisions

Were a revised manuscript to be sent for another round of reviews:

I would be willing to review the revised manuscript.

We appreciate your review and critique of the manuscript. Thank you.

Dear authors, many thanks for revising the manuscript. It looks like many of my comments were
addressed and the reading is now more clear and fluid.

I still have one main comment about this work, which arises from the Results section and the
main goal of the manuscript.

I appreciate the fact that you introduced more Z-S relationships in your fig.12 plot (Hiley et al,
SSKB), it really helps contextualizing your work. The comparison in the result section though is
a bit confused and it jumps all over the places, mentioning Hiley et al, but then describing
Matrosov to suddenly go back to PV11 (describing the cut at $Z>10mm6m-3$ and the best fit line).
Then back to Hiley with a mention to SSKB (but without really comparing SSKB) and then
numbers to quantify the departure from Hiley and so far so on. All this to say that it would be
great to get a systematic description/comparison between the Z-S relationships obtained in this
work and all the others, starting for example from Matrosov, describing what kind of particles
they consider and why there is this departure from the points measured in this work and so on.
Also, SSKB looks like the best curve, the one that most resembles the points obtained in this
work, but it is not described or emphasized. I think it should take some more space in the
comparison. Also, the 4 points definitely fit within the PV11 variability and I think it would be
safe to say that you confirmed the PV11 relationship with measurements (at least within its
range, which is not just the best fit line). I think this might be the main goal of the manuscript as
far as I can read from what you presented.

Sect. 4 of the manuscript was rewritten. Please see revision2-Sect.4.

Finally, even if the goal of this manuscript is not to provide a rimed particle Z-S relationship, it would be very nice if you could list the four relationships you obtained.

S/Z relationships, formulated algebraically, are in Sect. 1. The formulas are graphed in revision2

(Figs. 12a – b). In the Figs. 12a – b caption, and in discussion of Figs. 12a – b (revision2-Sect.4), we refer the reader to Sect. 1 where the formulas are provided. It is our opinion that this is sufficient.

Other comments:

l.21: Add "is" in "it IS also shown".

This is corrected in revision2.

l.105: Probably you mean Fig.12?

We meant Fig. 11.  That figure shows the fit ("S($\rho_1$)/Z best-fit line") and the S/Z pairs.  The sentence was revised. The revision2 text, and the prior sentence (unchanged), is provided here:

"In addition to variance in their values of S, coming from a dependence on density, PV11 state that a value of S derived via their best-fit line is uncertain by a factor-of-ten. That uncertainty is evident in the variance of S/Z data pairs about the line in Fig. 11 of PV11."

l.133: it would be useful for contextualization to add here that WKA was flying from Laramie to

Saratoga (if I understood it correctly from the response to reviewers) as test flights in preparation to SNOWIE. I see this mentioned in l.182, so not a big deal, but I feel like it would be better to contextualize it earlier on (so around l.133 when you mention the overflights or at the beginning of section 2.2 where you introduce the WKA).

 The statement was moved to revision2-Sect.2.1 where it is stated that the "…flights…were flown from the Laramie, WY airport…" We hope this revision/change does _not_ leave the impression that the overpasses were flown east to west. In fact, the overpasses were flown west to east (upwind to downwind). This is stated in revision2-Sect.3.1.

L.203: "The latter was not corrected for snow particle undercatch; however, in what follows we describe that correction" – I am not sure I fully understand this sentence. With "the latter" do you mean the liquid-equivalent snowfall rate? Is it or is it not corrected for undercatchment? If there is a correction method ("in what follows we describe that correction") why isn't it applied? Please clarify this sentence to make clear if a correction has been applied (if not, why bothering describing it? If relevant it should be explained why it was not applied).

The offending paragraph, the paragraph that proceeds, and the paragraph that follows, were revised. Please see revision2-Sect. 2.4.

L.276: I assume this note is just for the manuscript and the table will be put around here in the final paper?

This is fixed in revision2.

l.342-346: I still think that mentioning this comparison does not provide any useful information.

We removed this. We also removed the Vaisala (2012) citation.

l.597: Here you mention the $i=0$ and $i=1$ intervals while in the figures I see $i=0$ and $i=2$. I haven't seen the $i=2$ while you were introducing the intervals in fig.6 and paragraph from l.510 to 514 where it looks like you are introducing the intervals for the first time. Maybe you want to introduce $i=0$, $i=1$ and $i=2$ at the beginning (paragraph 510-514)? Also fig. 8 shows two intervals, fig.7 only one, is there a reason why? Try to be consistent and if this is the way the analysis needs to be done then just specify the 3 intervals from the beginning so the reader doesn't wonder what $i=2$ is for most of the section before finding the explanation.

This was corrected in revision2-Sect.3.5.

L.998: the link to Marlow, S.A, J.M. Frank, M. Burkhart, B. Borkhuu, S.E. Fuller, and J.R.

Snider, Snowfall measurements in mountainous terrain, in revision for the Journal of Applied

Meteorology and Climatology, http://www-das.uwyo.edu/~jsnider/JAMC-D-22-0093_6.pdf,

2023 1000 does not work and it is not possible to check this reference (which is heavily used for methodology explanation).

We took down (from the web) our first submission to JAMC. We did that after responding to critiques and producing a revised manuscript. However, we did not post the revision. We apologize.

The citation is corrected, and the link is available, in revision2 of this paper:

Marlow, S.A, J.M. Frank, M. Burkhart, B. Borkhuu, S.E. Fuller, and J.R. Snider, Snowfall

Measurements at Wind-exposed and Sheltered Sites in the Rocky Mountains of Southeastern

Wyoming, in revision for the Journal of Applied Meteorology and Climatology, http://www- das.uwyo.edu/~jsnider/manuscript_revision2.docx, 2023

---

## Author Response (AR3)

For final publication, the manuscript should be
accepted subject to minor revisions

Were a revised manuscript to be sent for another round of reviews:
I would be willing to review the revised manuscript.

We appreciate your review and critique of the manuscript. Thank you.

Dear authors,

Thanks again for your revised version of the manuscript. I went through it once again and now the results section reads much better and provides a fluid description of the Z-S relationships and how yours compare.

I have a few last suggestions:

1. I appreciate the listed Z-S relationships in section one, but I would also add a table in section 4 (which you can reference from section 1 as well) with all the relationships, their acronyms, what particles have been used and what is the departure of your points. I see that you calculated the departure only from Hiley (upper limit), why is that? Why didn't you calculate the departure from all the suggested relationships for a better comparison? I would suggest all the departures in section 4 and add the results in a table.

A table was added. Table 1, in revision3, incorporates all your suggestions. The last two columns of the table have the minimum relative S differences. Please note that the differences are calculated with an absolute value (|(SHP-S)|/S), while in revision2, the differences were calculated without the absolute value ((SHP-S)/S). The S differences are discussed in Sections 4 (Results) and 5 (Conclusions).

2. As I suggested you mentioned something about SSKB being closer to your points (lines 854-855), but you are missing the why. Is it because of the density? Because of the spherical shape assumption?

Since two things are different in SSKB (density and shape), compared to Hiley et al., an answer to your question must be accompanied by uncertainty and discussion. We did that in Sections 4 of revision3. Here is the added text:

"Figure 12 compares our $S_{HP}/Z'$ data pairs to the SSKB S/Z relationship line and Table 1 presents the relative differences between the data pairs and the SSKB line. Compared to the S/Z relationship represented by the top of the orange region in Fig. 12, and compared to the Matrosov 2007 relationship, the SSKB line plots closer to our data points (minimum relative difference ~ 0.3). We note that the only instances of $S_{HP} < S$ are three of four comparisons of our measurements to the SSKB relationship. A possible reason for this is that the density applied in SSKB (Table 1) is not entirely representative of conditions during our study. An analysis of the sensitivity of the SSKB to a change in density is needed to investigate our assertion."

Can you expand this 2 lines sentence you added? Why focusing so much on Hiley calculating departure, errors etc. when it is obviously underestimating because it does not consider the right particles.

We focused on Hiley et al. because those researchers provided an S/Z relationship applied in retrievals of S based on W-band measurements. In revision3, Sect. 4, we expanded on this issue:

"In comparisons of our snowfall rates and the upper-limit S/Z relationship line from Hiley et al. (2011) the relative difference is no smaller than 0.7 and 1.0 on 15 December and 3 January, respectively. These minimum relative differences exceed the hotplate precision (Sect. 2.4) by at least a factor of two. It is concluded that our paired values of undercatch-corrected precipitation rate and attenuation-corrected radar reflectivity provide evidence that a calculation of S based on the Hiley et al. (2011) upper-limit, when applied to rimed snow particles, is associated with a low-biased estimate of S. A retrieval based on Hiley et al.'s average S/Z relationship (not shown), which bisects the orange region in Fig. 12, corresponds to an even larger low bias. This is a concern because Hiley et al. (2011) used their average S/Z relationship to retrieve global snowfall distributions and since global observations reported in Wang et al. (2013) document the frequent occurrence of supercooled liquid within snowing clouds."

It is good to report all the information you provided about Hiley, but I still think it is important to give the same amount of space to relationship that are closer to your points (like SSKB or PV11).

Comparable space is given to SSKB, PV11, and Falconi et al. in Section 4 of revision3 (Results).

3. Why introducing a new plot in figure 12? The plot has only Falconi et al. as new lines, Matrosov and Hiley are repeated. Please consolidate in one plot for easier comparison with your points (not reported in the second plot, how are we supposed to compare?) and maybe put the legend outside the plot space.

In revision3 we redrew Fig. 12 with one panel and with the legend above the plot space.

4. In section 3.5 you introduce the intervals i=0, i=1 and i=2. You should include i=1 in the figures as well (figs. 7 to 10) since it is one of the two points you show in your results in fig. 12. For sure i=1 points are more important to show than i=2 since the i=1 are a crucial part of your analysis and in the figures it is not obvious where they are located (especially because they anticipate t0 as opposed to the ground data that follow). I think showing i=0 and i=2 only creates confusion on what is actually important to focus on.

We think the Figs. 6 schematic of the i=0 and i=1 hotplate averaging intervals and of the i=0 and i=1 WCR averaging intervals is adequate for informing the reader that the i=0 and i=1 intervals are what's most important. That said, we agree that an improvement is to indicate both the i=0 and i=1 in Figures 7, 8, 9, and 10.  That change was implemented in the revision3.

Also, we stated (L548 in revision2) that Fig. 8 shows the i = 2 averaging interval as a "special case" and that this is discussed at the end of Sect. 3.5. It is also stated, at the end of the following paragraph (L559 in revision2), after introducing the airborne radar measurements (Figs. 10a-b), that the i = 2 interval is shown and that the reason for this is discussed at the end of Sect. 3.5.

The statements about the i = 2 averaging interval, from revision2, remain in revision3.

5. In table 5 can you also add a column with attenuation corrected Z? So we have an idea on the values you are actually showing in the plot (fig.12) since those are attenuation corrected values.

Attenuation-corrected values were added to the final column of Table 5.

6. line 780 you mention 'summit' – do you mean the ridge line?

Yes, that is what we meant.  For clarity, the text in revision3 was changed to this:

"First, the two snowfall events had flight-level vertical wind velocities (Figs. 5b and 5d) that were positive (upward) upwind of the ridgeline, and vice versa downwind of the ridgeline.

Except for the strongest downdraft on 3 January 2017, the magnitude of this variance is ~ 1 m s$^{-1}$ (Figs. 5b and 5d)."

If that's the case can you add an indicator as you did in fig. 5 so it is more obvious where to look?

In revision3, a "Ridgeline" indicator is added to the panels showing flight-level measurements of vertical velocity (Figs. 5b and 5d).

7. In the conclusions can you also add some summary of your findings about SSKB and Falconi? You still only focus on PV11 and Hiley and disregarded some relationships that are closer to your results.

Comparable space is given to SSKB and Falconi et al. in the Section 5 (Conclusions) of revision3.

8. I see there are really a few (3) references from 2020, isn't there anything more recent to reference? Most of the references are at least 10 years old.

Two post-2019 references were added to revision3. In revision3, there are five post-2019 references. This count does not include MS theses or web links to data sets.

The post-2019 references added to revision3 are these:

Battaglia, A., Tanelli, S., Tridon, F., Kneifel, S., Leinonen, J., and Kollias, P, Triple-Frequency Radar Retrievals, In: Levizzani, V., Kidd, C., Kirschbaum, D.B., Kummerow, C.D., Nakamura, K., Turk, F.J. (eds) *Satellite Precipitation Measurement*, Advances in Global Change Research, vol 67. Springer, Cham. https://doi.org/10.1007/978-3-030-24568-9_13, 2020

Vogl, T., Maahn, M., Kneifel, S., Schimmel, W., Moisseev, D., and Kalesse-Los, H.: Using artificial neural networks to predict riming from Doppler cloud radar observations, Atmos. Meas. Tech., 15, 365–381, https://doi.org/10.5194/amt-15-365-2022, 2022

---

## Author Response (AR4)

**The following pages have versions of Sect. 5 submitted to AMT on 10/12/2023 and on**

**11/02/2023.**

**J.Snider**

**11/02/2023**

**Here is the version of Sect. 5 submitted to AMT 10/12/2023:**

[revised manuscript text omitted]